# The integrated stress response drives *MET* oncogene overexpression in cancers

Marina Cerqua [ID][1], Marco Foiani[1], Carla Boccaccio [ID][2,3], Paolo M Comoglio [ID][1,4 ✉] &
Dogus M Altintas [ID][1,4 ✉]

## Abstract

**Cancer cells rely on invasive growth to survive in a hostile micro-environment; this growth is characterised by interconnected processes such as epithelial-to-mesenchymal transition and migration. A master regulator of these events is the MET oncogene, which is overexpressed in the majority of cancers; however, since mutations in the MET oncogene are seen only rarely in cancers and are relatively infrequent, the mechanisms that cause this widespread MET overexpression remain obscure. Here, we show that the 5′ untranslated region (5′UTR) of MET mRNA harbours two functional stress-responsive elements, conferring translational regulation by the integrated stress response (ISR), regulated by phosphorylation of eukaryotic translation initiation factor 2 alpha (eIF2α) at serine 52. ISR activation by serum starvation, leucine deprivation, hypoxia, irradiation, thapsigargin or gemcitabine is followed by MET protein overexpression. We mechanistically link MET translation to the ISR by (i) mutation of the two uORFs within the MET 5′UTR, (ii) CRISPR/Cas9-mediated mutation of eIF2α (S52A), or (iii) the application of ISR pathway inhibitors. All of these interventions reduce stress-induced MET overexpression. Finally, we show that blocking stress-induced MET translation blunts MET-dependent invasive growth. These findings indicate that upregulation of the MET oncogene is a functional requirement linking integrated stress response to cancer progression.**

**Keywords** MET Oncogene; Integrated Stress Response; Invasive Growth
**Subject Categories** Cancer; Translation & Protein Quality

## Introduction

Cancer cells thrive in a harsh microenvironment characterised by limited nutrients, hypoxia, toxic waste, inflammation, and exposure to cytotoxic treatments (Anderson and Simon, 2020; Parker et al, 2021). Despite neo-angiogenesis, the supply of metabolites and oxygen, as well as clearance of toxic products, are impaired due to the disorganised architecture of newly formed vessels (Harris, 2002;

Höckel and Vaupel, 2001; Lin et al, 2016; Vaupel et al, 1989). Cancer cells must activate a prompt and robust stress response to survive in this challenging milieu. The *integrated stress response* (ISR) is an evolutionarily conserved cellular defence mechanism that enables cells to adapt their protein synthesis in response to various hostile challenges (Cigan et al, 1991; Dever et al, 1992; Garcia et al, 2007; Harding et al, 2000; Harding et al, 2003; Pakos-Zebrucka et al, 2016). Through the phosphorylation of eukaryotic translation initiation factor 2 alpha (eIF2α) at serine 52, performed by the four serine/threonine kinases—General Control non-derepressible 2 (GCN2); protein kinase R-like endoplasmic reticulum kinase (PERK); Haem-Regulated eIF2α kinase (HRI); or Double-stranded RNA-dependent protein kinase (PKR)—the ISR suppresses general protein synthesis while selectively promoting the translation of mRNAs encoding proteins crucial for cell survival (Dever et al, 1992; Pakos-Zebrucka et al, 2016). These transcripts contain upstream open reading frames (uORFs) in their 5′ untranslated region (5′UTR) (Pakos-Zebrucka et al, 2016). Specifically, eIF2α phosphorylation promotes leaky ribosomal scanning, which allows bypassing the translation of uORFs and, therefore, preferentially enhances the translation of the main coding sequence (Pakos-Zebrucka et al, 2016). One of the best-known ISR translational target is ATF4, a transcription factor controlling the expression of genes involved in cell survival (Harding et al, 2003; Kaufman et al, 2002; Vattem and Wek, 2004). Notably, ATF4 is frequently overexpressed in various cancer types (Elefteriou et al, 2006; González-González et al, 2018; Qing et al, 2012), suggesting the reliance of tumour cells on ISR for survival (Costa-Mattioli and Walter, 2020). However, the role of ISR in tumour pathogenesis remains elusive.

Adverse environmental conditions are known to activate invasive growth, a complex series of processes that encompass cell dissociation, migration, growth, and survival, ultimately promoting cancer progression (Birchmeier et al, 2003; Boccaccio and Comoglio, 2006; Comoglio et al, 2018). The *MET* oncogene is a master driver of invasive growth (Comoglio et al, 2018; Lee et al, 2016). It is pervasively overexpressed in over 70% of cancers, making it one of the top five oncogenes prioritised for targeted therapies (Behan et al, 2019). MET activation by its ligand, hepatocyte growth factor (HGF), triggers a cascade of downstream signalling pathways, including the PI3K/AKT, RAS/MAPK, and STAT3 pathways, which are essential for driving oncogenesis,

[1]IFOM ETS—The AIRC Institute of Molecular Oncology, 20139 Milano, Italy. [2]Candiolo Cancer Institute, 10060 Candiolo, Torino, Italy. [3]Department of Oncology, University of Torino, 10100 Torino, Italy. [4]These authors contributed equally: Paolo M Comoglio, Dogus M Altintas. ✉E-mail: paolo.comoglio@ifom.eu; dogus.altintas@ifom.eu

tumour progression, and metastasis (Reviewed in: (Birchmeier et al, 2003; Boccaccio and Comoglio, 2006; Comoglio et al, 2018)). Furthermore, *MET* is known to be upregulated in response to environmental stresses such as hypoxia (Pennacchietti et al, 2003) or irradiation (De Bacco et al, 2011), highlighting its role in adaptive resistance mechanisms that contribute to therapeutic challenges in cancer treatment. However, the transcriptional control by factors such as hypoxia-inducible factor or NF-ƙB can only partially explain the pervasive overexpression of *MET* in cancer, highlighting the need for additional mechanisms supporting the *MET* oncogene inducible nature (De Bacco et al, 2011; Pennacchietti et al, 2003). Here, we show that *MET* mRNA contains two uORFs, critical in upregulating *MET* translation in response to various microenvironmental challenges. We found that inhibiting ISR through either genetic or pharmacological interventions curtailed such upregulation. Hampering ISR affected the MET-driven invasive growth of cancer cells, as it blunted epithelial-to-mesenchymal transition, wound healing, growth, and survival. This study highlights a new level of *MET* oncogene expression control in cancer and establishes a pivotal mechanistic connection between the integrated stress response pathway and invasive growth.

## Results

### MET mRNA translation is upregulated by a broad range of stress stimuli through the 5'UTR

A defining feature of transcripts regulated by ISR, including the well-documented *ATF4*, is the presence of upstream open reading frames (uORFs) within their 5' untranslated regions (5'UTRs) (Pakos-Zebrucka et al, 2016). Through database analysis and building on published data, we found that the *MET* and *ATF4* 5'UTRs share notable similarities in length (396 bases and 282 bases, respectively) and the presence of two uORFs (Fig. 1A). In *ATF4*, uORF1 encodes a 3 amino acid peptide, and uORF2 encodes a 59 amino acid peptide; both are out of frame with the main ORF, with uORF2 overlapping the main ORF (Vattem and Wek, 2004). Similarly, uORF1 of *MET* encodes a 4 amino acid peptide in-frame with the main ORF and uORF2 encodes a 16 amino acid peptide out of frame, neither overlapping with the main ORF (Wethmar et al, 2016; Yang et al, 2021). The structural parallels between *MET* and *ATF4* 5'UTRs raise the possibility of an ISR-mediated regulation of *MET* translation, similar to that of *ATF4*.

To investigate this hypothesis, we utilised a reporter system, cloning the *MET* 5'UTR upstream of a Firefly luciferase gene (Fluc), with a CMV promoter-driven Renilla luciferase vector used as a normaliser (Fig. 1B). The effect of the 5'UTR on reporter expression was assessed by normalising to a No-5'UTR control. At a steady state, the *MET* 5'UTR decreased Fluc expression as compared to no 5'UTR. When cells were challenged with different stimuli known to activate ISR [serum starvation (Ye et al, 2010), chemotherapy (represented by Gemcitabine, GEM (Palam et al, 2015)), irradiation (Collier et al, 2015; Pakos-Zebrucka et al, 2016), thapsigargin (Harding et al, 2000), hypoxia (Koumenis et al, 2002), or leucine deprivation (Harding et al, 2000)], Firefly luciferase signal significantly increased (Fig. 1C). Mutations in either uORF1 or uORF2 diminished this increase, while the double mutant (uORF1/2) abolished it, highlighting the critical

role of both uORFs in stress responsiveness. As expected, the Renilla luciferase signal remained unchanged (Fig. 1D). Superimposable results were obtained using a GFP reporter system (Fig. EV1A–C). In addition to uORFs, internal ribosomal entry sites (IRES) are known to mediate translation during stress in a cap-independent manner (Spriggs et al, 2008). Using RNAfold (Lorenz et al, 2011) and the Human IRES Atlas (Yang et al, 2021), we predicted the presence of a stable structure within the *MET* 5'UTR (302–396 bases) that might function as an IRES (Fig. EV2A,B). To test its functionality, we cloned the putative IRES region between Renilla and Firefly luciferase genes in a bicistronic vector. A known poliovirus IRES served as a positive control (Pelletier and Sonenberg, 1988), while promoterless vectors were used as negative controls (Fig. EV2C). However, the predicted *MET* IRES showed no significant activity, as indicated by the lack of Firefly signal (Fig. EV2D). Renilla signal validated the functionality of the bicistronic system (Fig. EV2E). Altogether, these results suggest that the stress-inducible nature of the *MET* 5'UTR is driven by uORFs rather than IRES.

The stress-induced upregulation of endogenous MET protein was consistently observed in EKVX lung adenocarcinoma cell lines challenged with various stressors. Whether exposed to serum starvation, hypoxia, irradiation, thapsigargin, GEM, or leucine deprivation, MET protein levels increased with kinetics and magnitude comparable to the ISR marker ATF4 (Fig. 1E). Complementary experiments in an isogenic EKVX *MET* knockout cell line, with reintroduction of the *MET* coding sequence ± 5'UTR, further demonstrated the role of the 5'UTR in stress-induced MET expression (Fig. 1F). In summary, our findings illuminate the unsuspected role of the *MET* mRNA 5'UTR in supporting a significant increase of the MET protein in cells undergoing ISR.

### MET translation and the subsequent biological activity are regulated by the two uORFs

We next used prime editing (Anzalone et al, 2019; Chen et al, 2021) to mutate the two uORFs of *MET*, investigating their necessity for stress-induced upregulation in comparison to an isogenic *MET* wild-type (wt) cell line. Serum starvation triggered the ISR in both wt and MET uORF1/2-edited EKVX cells (mutations in uORF1 and uORF2), as evidenced by the activation markers: phosphorylation of eIF2α, increased ATF4 expression, and upregulation of ATF4-regulated genes such as *ATF3*, *GADD45A*, *GADD45B*, and *DDIT3* (Ahola et al, 2022) (Figs. 2A and EV3A). In wt cells, ISR activation increased MET protein levels. However, mutations in *MET* uORF1/2 reversed this effect, transforming *MET* into a "non-stress regulated" gene. While wt MET levels rose under stress, uORF1/2 mutant cells showed a sharp decrease in MET protein levels following serum withdrawal. This mirrored response underscores the role of uORFs in maintaining stress responsiveness, as non-stress-regulated transcripts typically decrease during ISR activation (Pakos-Zebrucka et al, 2016). Serum starvation did not affect MET mRNA levels, confirming the post-transcriptional regulatory role of the uORFs, similar to ATF4 (Fig. 2B). In addition, the phosphorylation of MET in response to HGF, and the ensuing signal transduction (indicated by AKT phosphorylation), were consistent with the differences in MET protein levels between wt and uORF1/2 mutant cells (Fig. 2A).

The HGF/MET signalling cascade triggers various biological responses collectively known as invasive growth, including epithelial-to-mesenchymal transition (EMT), migration, survival/growth, and

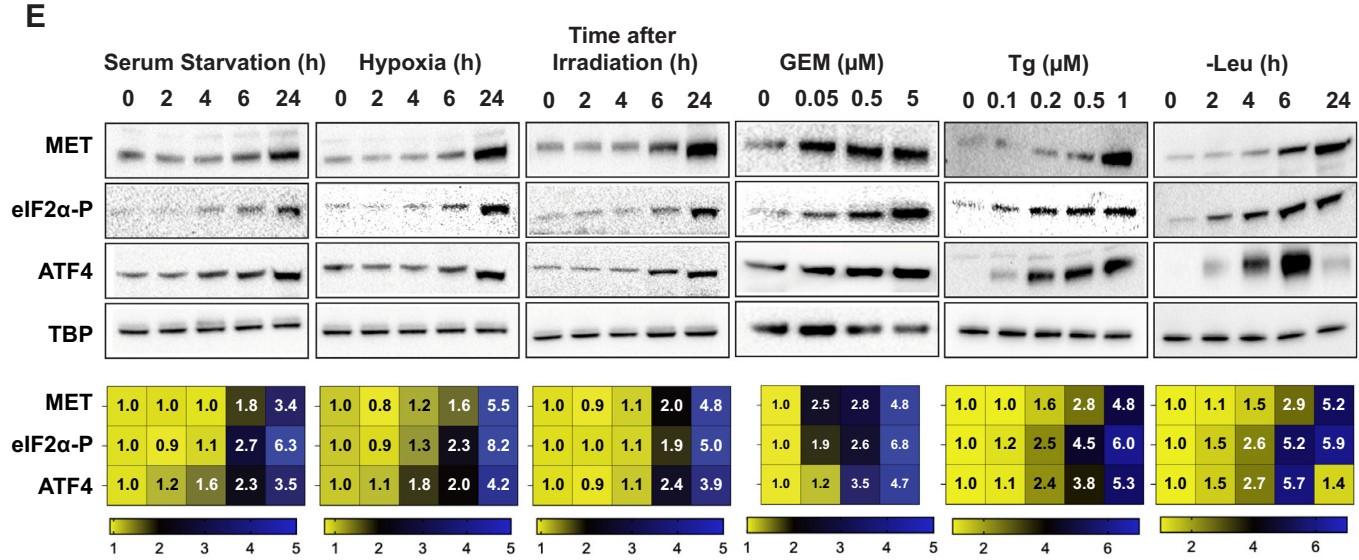

**A**

MET 5'UTR schematic showing TSS, uORF1 (99–110), uORF2 (250–297), Start codon AUG (396), Exon 1, Exon 2, CDS.

**B**

Luciferase reporter constructs: No 5'UTR, MET 5'UTR

| | |
|---|---|
| AUGcccggcUGA AUGcggagccgaguggagggcgcgagccagaugcggggcgacagcUGA | Wild Type (wt) |
| AAGcccggcUGA AUGcggagccgaguggagggcgcgagccagaugcggggcgacagcUGA | uORF1 mut |
| AUGcccggcUGA UUCcggagccgaguggagggcgcgagccagaugcggggcgacagcUGA | uORF2 mut |
| AAGcccggcUGA UUCcggagccgaguggagggcgcgagccagaugcggggcgacagcUGA | uORF1/2 mut |

**C** Firefly

**D** Renilla — Row factor: p=0.052, Column factor: p=0.8865

Legend: Steady State, Serum Starvation, GEM, Irradiation, Hypoxia, Tg, -Leu

**F**

Western blot: 5'UTR (+ + - -), Serum (+ - + -), Control; MET (P, M), TBP

**E**

Western blots for Serum Starvation (h), Hypoxia (h), Time after Irradiation (h), GEM (μM), Tg (μM), -Leu (h). Proteins: MET, eIF2α-P, ATF4, TBP. Heatmaps below.

**Figure 1.    *MET* mRNA translation is upregulated by a broad range of stress stimuli through the 5'UTR.**

(A) Schematic representation of the *MET* mRNA, indicating the location of the two upstream open reading frames (uORF1 and uORF2) within the 5' untranslated region (5'UTR) upstream of the *MET* coding sequence (CDS). (B) Diagram of the firefly luciferase reporter construct used to assess the activity of *MET* 5'UTR. Wild-type (wt) and mutant (mut) versions of the uORFs are shown, with mutations highlighted in red in the sequence comparison. (C) Quantitative analysis of Firefly luciferase activity normalised to no UTR conditions (termed as control), demonstrating the effects of *MET* 5'UTR wt and mutant uORFs on translation after 24 h stress conditions (or steady state), including serum starvation, treatment with 500 nM gemcitabine (GEM), 10 Gray irradiation, hypoxia (3% oxygen), 1 µM thapsigargin (Tg) treatment, and Leucine deprivation (-Leu). Error bars represent mean ± SEM with $n = 6$ biological replicates. The P value is calculated using two-way ANOVA with Tukey's post hoc test and represents the row factor (different constructs, $P < 2.10^{-16}$) and column factor (different treatments, $P < 2.10^{-16}$). Exact P values for key comparisons: Control (steady state) vs. Control (GEM), $P = 1$; MET 5'UTR wt (steady state) vs. MET 5'UTR wt (GEM), $P = 2.96 \times 10^{-16}$; uORF1 mut (steady state) vs. uORF1 mut (GEM), $P = 1.75 \times 10^{-14}$; uORF2 mut (steady state) vs. uORF2 mut (GEM), $P = 1.62 \times 10^{-14}$; uORF1/2 mut (steady state) vs. uORF1/2 mut (GEM), $P = 1$; Control (Irradiation) vs. MET 5'UTR wt (Irradiation), $P = 2.03 \times 10^{-6}$; MET 5'UTR wt (Irradiation) vs. uORF1 mut (irradiation), $P = 6.45 \times 10^{-9}$; uORF1 mut (irradiation) vs. uORF2 mut (irradiation), $P = 6.48 \times 10^{-9}$; uORF2 mut (irradiation) vs. uORF1/2 mut (irradiation), $P = 7.72 \times 10^{-12}$. (D) Renilla luciferase control measurements across the same conditions as in (C). Error bars represent mean ± SEM ($n = 6$ biological replicates), and statistical analysis was performed as in (C). (E) Time-course (in hours) western blot analysis showing the induction of MET protein, phosphorylation of eIF2α (eIF2α-P), and ATF4 expression in response to serum starvation, hypoxia, irradiation, Leucine deprivation, as well as a dose-dependent increase in MET expression upon treatment with GEM (in µM) for 24 h or Tg (in µM) for 6 h. TATA-binding protein (TBP) serves as a loading control for each condition. Heatmaps show relative band intensities with a numbered scale, where yellow indicates low levels and blue indicates high levels. (F) EKVX *MET* KO cells were infected with a lentivirus containing the full *MET* CDS containing or not the 5'UTR. Cells were grown in ± serum for 24 h. "M" denotes the mature form of the receptor, "P" stands for the precursor (Komada et al, 1993). The blot confirms the significance of the 5'UTR in enhancing MET protein expression under serum starvation, with TBP as a loading control. Significance markers: *$P < 0.05$, **$P < 0.01$, ***$P < 0.001$, ****$P < 0.0001$, "ns" indicates not significant. Source data are available online for this figure.

protection from apoptosis (Birchmeier et al, 2003; Boccaccio and Comoglio, 2006). To highlight the importance of uORFs in invasive growth, we conducted a series of experiments. Consistent with stress-induced MET upregulation, HGF effectively triggered EMT under serum starvation in wild-type cells, as evidenced by the upregulation of mesenchymal genes *ACTA2*, *SNAI1*, *SNAI2*, and *VIM*, along with the downregulation of epithelial genes *CDH1*, *CDH2*, *TJP1*, and *TJP2* (Fig. 2C). This HGF-dependent EMT induction was reduced when wt cells were cultured in the presence of serum, highlighting the relevance of stress-induced MET upregulation for these responses. In sharp contrast, MET uORF1/2 mutant cells exhibited a markedly different EMT response. While HGF only mildly induced EMT-related gene expression under serum starvation, it significantly enhanced EMT when these cells were grown in serum-rich conditions, consistent with the altered MET protein levels in uORF1/2 mutant cells. We next tested other features of invasive growth: survival/growth and inhibition of apoptosis. HGF enhanced the viability of wt cells under serum starvation, but this effect was significantly blunted in uORF1/2 mutant cells (Fig. 2D). When cells were subjected to Gemcitabine (GEM), another stressor known to activate ISR (Hayashi et al, 2011; Palam et al, 2015), HGF treatment resulted in an eightfold increase in IC$_{50}$, in line with MET's recognised anti-apoptotic function (Boccaccio and Comoglio, 2006; Xiao et al, 2001). This protective effect was significantly compromised in MET uORF1/2 mutant cells (Fig. 2E). No difference in cell viability between wild-type and uORF1/2 mutant cells was observed under control conditions, ruling out any off-target effect of prime editing (Fig. EV3B).

Collectively, these results underscore the essential role of *MET* mRNA uORFs in modulating MET expression and subsequent signalling activation under stress conditions that activate ISR.

## MET expression and the ensuing biological activity are controlled by the *integrated stress response* regulator eIF2α

Next, we focused on the phosphorylation of eIF2α, the crossroad of the ISR response (Fig. 3A and (Harding et al, 1999; Pakos-Zebrucka et al, 2016)). To investigate its role in MET regulation, we treated cells with Salubrinal (SAL), an inhibitor of GADD34 that prevents the dephosphorylation of eIF2α (Boyce et al, 2005). Increased concentration of SAL treatment enhanced eIF2α phosphorylation as well as ATF4 protein levels (Fig. 3B). Mild but consistent upregulation of MET protein levels were observed between 1 µM and 10 µM of SAL without any corresponding increase in *MET* mRNA levels (Fig. 3B,C). This observation aligns with the re-initiation model known in other uORF-containing mRNAs, such as *ATF4* (Boyce et al, 2005; Harding et al, 2000; Vattem and Wek, 2004). In addition, SAL significantly increased the mRNA levels of known ATF4 target genes (Fig. EV4A).

To complement these findings, we set out to genetically dissect the chain of events linking stress stimuli to the upregulation of *MET* translation. Using CRISPR-guided precise editing, we engineered isogenic knock-in EKVX cells to express a mutated form of eIF2α (S52A cells), lacking the residue undergoing regulatory phosphorylation. As expected, despite comparable total eIF2α protein levels between wt and S52A cells, no phosphorylated eIF2α was detected in the edited cells under serum starvation (Fig. 3D). The lack of ISR activation in S52A cells was further confirmed by the absence of ATF4 upregulation and the failure to induce ATF4 target genes (Figs. 3D and EV4B). While wt cells activated the ISR and upregulated MET protein under stress, S52A cells were unable to promote stress-dependent MET translation (Fig. 3D). Consistent with the lower MET protein levels in S52A cells, reduced HGF-induced MET phosphorylation and down-stream AKT activation were observed compared to wild-type cells (Fig. 3D). In all conditions, *MET* and *ATF4* mRNA levels remained unchanged, ruling out any transcriptional regulation (Fig. 3E).

The inability of S52A cells to activate the stress response and MET translation negatively impacted the biological response to HGF. Indeed, under serum starvation, S52A cells failed to increase cell viability in the presence of HGF (Fig. 3F). In another set of experiments, cells were treated with GEM. Also in this case, while HGF improved viability and increased GEM IC$_{50}$ by eightfold in parental cells, it failed to do so in S52A cells (Fig. 3G). In control conditions, S52A cell growth was comparable to the wt counter-parts (Fig. EV4C).

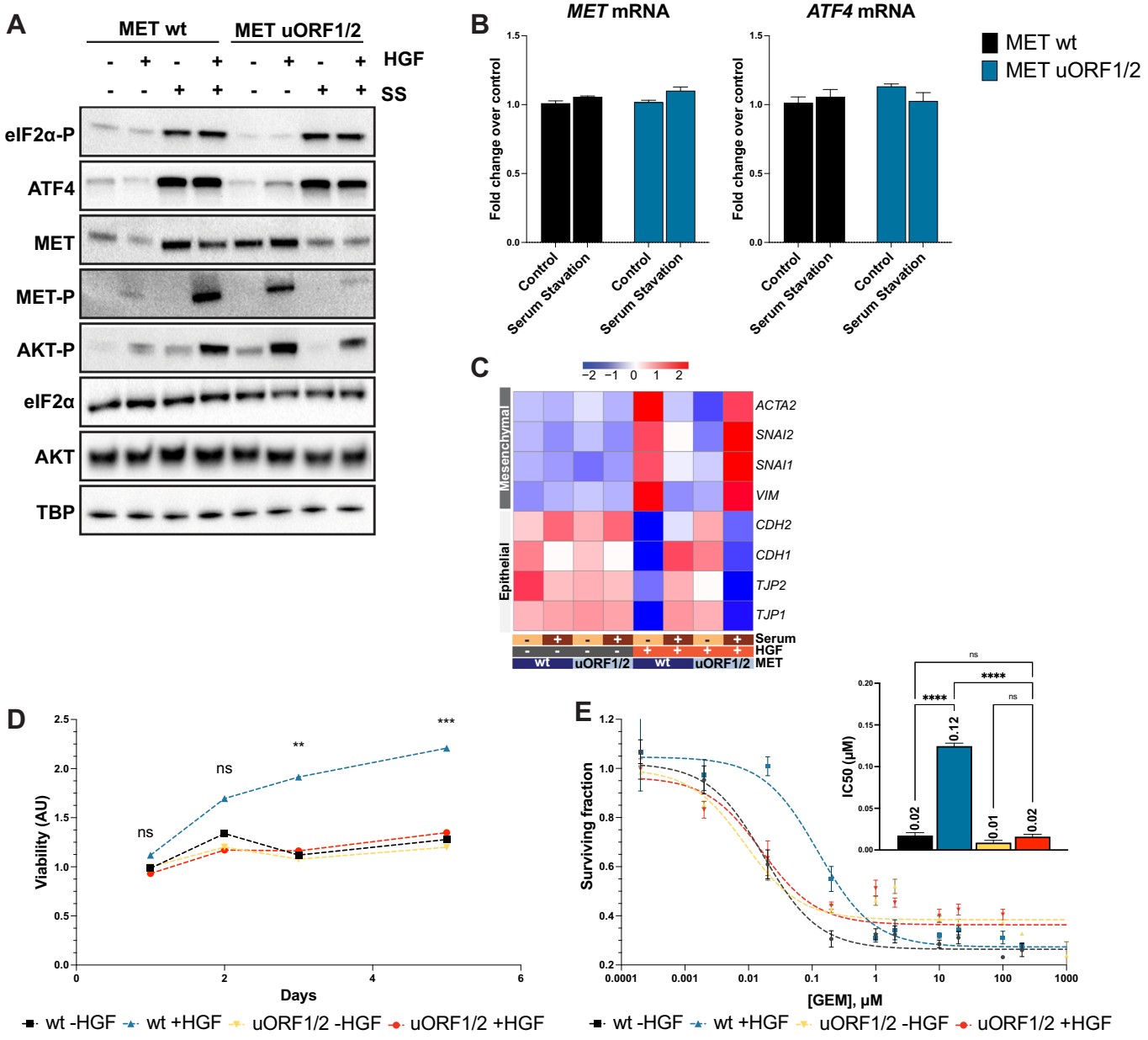

**Figure 2. *MET* translation and the subsequent biological activity are regulated by the two uORFs.**

(A) Western blot analysis demonstrating the impact of *M'T* 5′UTR through uORF1/2 on MET, MET-P, and AKT-P levels under control or under serum starvation (SS) for 24 h ± 50 ng/mL HGF during 15 min. eIF2α-P and ATF4 were used as ISR markers, with TBP as a loading control. (B) MET and ATF4 mRNA levels were assessed by RT-qPCR after mRNA extraction from cells ± serum starvation, using TBP as a normalisation gene. Results are presented as mean ± SEM ($n = 3$ biological replicates), with data analysed by one-way ANOVA and Tukey's post hoc test. (C) Heatmap showing the relative expression levels, assessed by RT-qPCR, of mesenchymal (*ACTA2*, *SNAI2*, *SNAI1*, *VIM*) and epithelial (*CDH2*, *CDH1*, *TJP2*, *TJP1*) marker genes in wt or uORF1/2-mutated EKVX cells ± serum ± 50 ng/mL HGF for 48 h. Gene expression levels are represented as the mean of $n = 3$ independent experiments, with a colour scale ranging from blue (downregulated) to red (upregulated). (D) Cell viability assay over time, showing the growth kinetics of cells with different genotypes, illustrating the influence of *MET* uORFs on cell survival under SS. Cells were treated with ± 50 ng/mL HGF, and results were normalised to fluorescence values measured at $t = 1$ day. Error bars represent SEM ($n > 6$ biological replicates), and data were analysed using two-way ANOVA with repeated measures and Tukey's post hoc test. (E) Dose–response experiments on cells with indicated genotype to increasing concentrations of GEM ± 50 ng/mL HGF for 48 h. Results were standardised to untreated conditions on wt cells, and $IC_{50}$ values were determined and represented on the inset. Error bars represent SEM ($n > 6$ biological replicates), with statistical analysis performed using one-way ANOVA and Tukey's post hoc test. Significance markers: $****P < 0.0001$, $***P < 0.001$, $**P < 0.01$, and "ns" indicates not significant. Source data are available online for this figure.

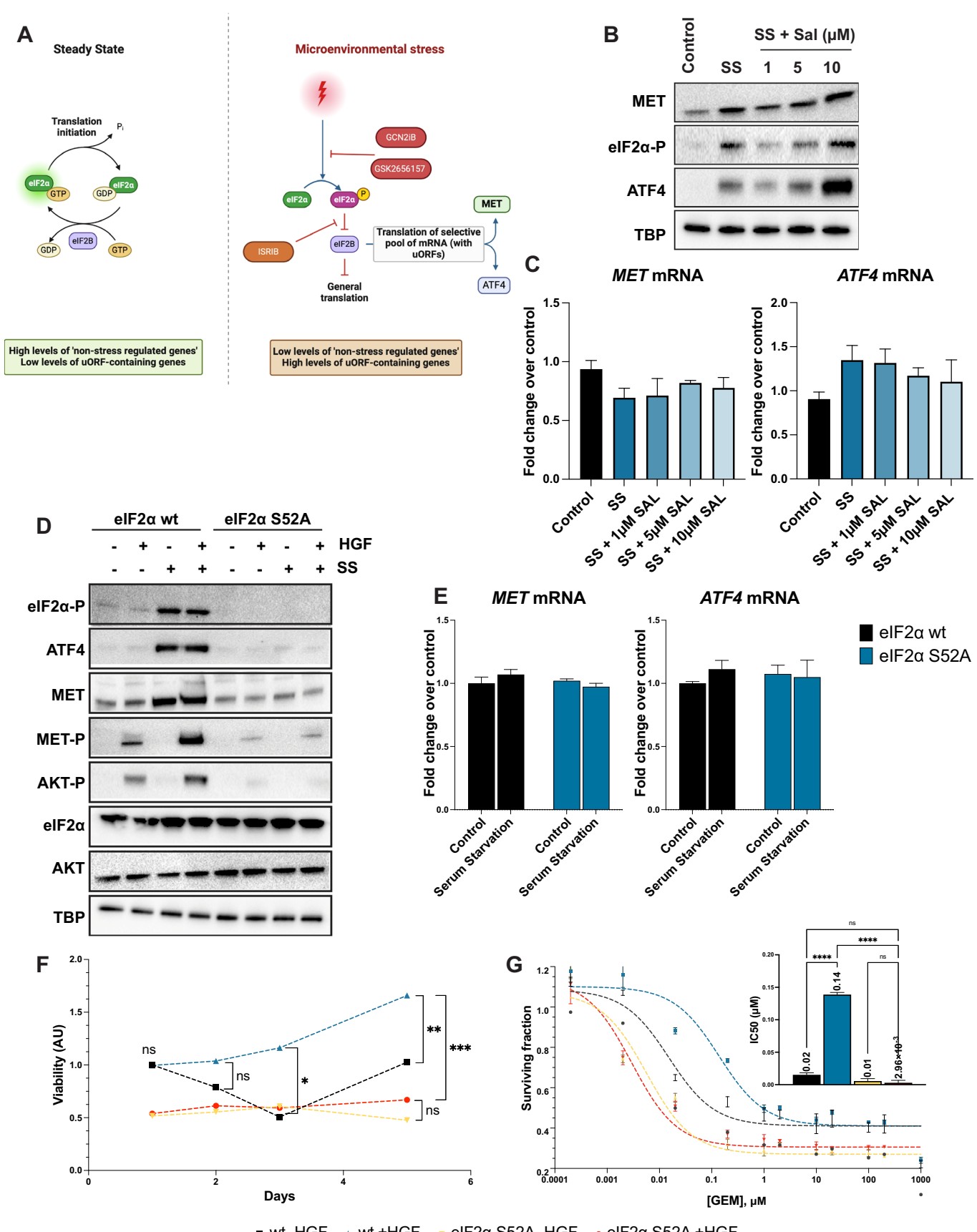

◀ **Figure 3. MET expression and the ensuing biological activity are controlled by the *integrated stress response* regulator eIF2α.**

(**A**) Schematic overview of translational control under steady-state and microenvironmental stress conditions. The diagram illustrates the transition from high levels of cap-dependent translation to preferential translation of stress-responsive genes via the integrated stress response-inducing eIF2α phosphorylation (Pakos-Zebrucka et al, 2016). The schematic also shows the action of three ISR inhibitors: GCN2 inhibitor GCN2iB (Nakamura et al, 2018), PER K inhibitor GSK2656157 (Atkins et al, 2013), and ISRIB (Rabouw et al, 2019; Zyryanova et al, 2021). (**B**) Western blot analysis showing the protein levels of MET, eIF2α-P, and ATF4 in cells subjected to 24 h of SS and treated with increasing concentrations of salubrinal (1, 5, and 10 μM). In control conditions, cells were grown in the presence of 10% serum without salubrinal. TBP was used as a loading control. (**C**) RT-qPCR analysis of *MET* and *ATF4* mRNA levels in cells treated as described in (**B**). mRNA levels are presented as fold change relative to the control condition, normalised to *TBP* expression. Error bars represent mean ± SEM ($n = 3$ biological replicates), with statistical analysis performed using one-way ANOVA and Tukey's post hoc test. (**D**) Western blot analysis comparing cells expressing the wild-type eIF2α or in CRISPR/Cas9 cells expressing non-phosphorylatable eIF2α S52A mutant. Phosphorylated eIF2α (eIF2α-P), ATF4, MET, phosphorylated MET (MET-P), and phosphorylated AKT (AKT-P) levels under normal and serum starvation (SS) conditions ±HGF (50 ng/mL), using TBP as a loading control. (**E**) Total RNA was extracted to assess *MET* and *ATF4* expression levels by RT-qPCR using *TBP* as a normalisation gene. Error bars represent mean ± SEM ($n = 3$ biological replicates), with statistical analysis performed using one-way ANOVA and Tukey's post hoc test. (**F**) Cell viability assay showing the proliferation profiles of cells with indicated genotypes, illustrating the impact of the eIF2α S52A mutation on cell survival during serum starvation. Cells were treated with or without 50 ng/mL HGF, and outcomes were normalised to fluorescence readings at $t = 1$ day. Error bars represent SEM ($n > 6$), and data were analysed using two-way ANOVA with repeated measures. (**G**) Dose–response experiments on cells with the indicated genotype to increasing concentrations of GEM ± 50 ng/mL HGF for 48 h. Results were standardised to untreated conditions in wt cells, and IC$_{50}$ values are shown in the inset. Error bars represent SEM ($n = 3$ biological replicates), with statistical analysis performed using one-way ANOVA and Tukey's post hoc test. Significance markers: ****$P < 0.0001$, ***$P < 0.001$, **$P < 0.01$, *$P < 0.05$, and "ns" indicates not significant. Source data are available online for this figure.

These results suggest that eIF2α phosphorylation is essential for the stress-induced increase in *MET* translation, and that such MET upregulation is biologically relevant under challenging conditions, including chemotherapy resistance.

## Stress-induced MET upregulation is impinged by ISR inhibition

We then inhibited the ISR/MET axis by targeting three distinct branches of the pathway with small molecule inhibitors: (i) the serine/threonine kinase GCN2, responsible for eIF2α phosphorylation, by the use of GCN2iB (Nakamura et al, 2018); (ii) the serine/threonine kinase PERK, responsible for eIF2α phosphorylation after endoplasmic reticulum (ER) stress, using GSK2656157 (Atkins et al, 2013); and (iii) the interaction between eIF2α and eIF2B, by the allosteric inhibitor ISRIB (Fig. 3A and Rabouw et al, 2019; Zyryanova et al, 2021).

To investigate the effects of GCN2 inhibition, we conducted dose–response experiments in EKVX cells under serum starvation (Nakamura et al, 2018). As serum concentration decreased, MET protein levels showed a progressive increase, comparably to well-established ISR markers such as phosphorylation of eIF2α and translation of *ATF4*. In total, 500 nM of GCN2iB effectively blocked stress-induced eIF2α phosphorylation and ATF4 synthesis, while also inhibiting MET protein upregulation (Fig. 4A). *MET* mRNA levels, measured by RT-qPCR, were not affected by GCN2iB, further supporting that MET control by ISR occurs at the translational level (Fig. 4B). mRNA levels of known ATF4 target genes were induced by ISR, but not in the presence of GCN2iB (Fig. EV5A). ISR-dependent MET protein upregulation was accompanied by increased MET phosphorylation and enhanced signal transduction (AKT phosphorylation) in the presence of the ligand HGF (Fig. EV5B). Importantly, MET protein levels and MET and AKT phosphorylation were not affected by GCN2iB under unstressed conditions, excluding any non-specific effects of the inhibitor on MET signalling (Fig. EV5B). Superimposable results were obtained under Thapsigargin-induced ER stress, where PERK inhibition with GSK2656157 effectively blocked ISR-induced MET upregulation (Fig. 4C), underscoring the key role of PERK in ISR-mediated MET regulation.

Interestingly, although *ATF4* mRNA levels were elevated during ER stress ((Martina et al, 2016) and Fig. EV5C), MET mRNA levels remained unchanged (Fig. 4D), suggesting that MET upregulation after PERK activation occurs primarily at a post-transcriptional level. We then hampered ISR with another drug, ISRIB, which acts as an allosteric inhibitor without affecting the phosphorylation status of eIF2α (Rabouw et al, 2019; Zyryanova et al, 2021). Dose–response experiments showed that ISRIB effectively inhibits *ATF4* and *MET* translation similarly to GCN2iB treatment (Fig. 4E,F). To demonstrate that ISR-induced *MET* translation and its targetability are not limited to a single cell line but are a general occurrence, we repeated these experiments in ten other cancer cell lines originating from various tissues (Figs. 4G and EV5D). All cell lines expressed comparable levels of wild-type MET without gene amplification (Fig. 4H), despite having diverse mutations or copy number variations in known cancer driver genes (Dietlein et al, 2020) (Fig. 4I).

These observations provide evidence that ISR-driven MET overexpression is a general mechanism, targetable by ISR inhibition, regardless of tissue origin and the presence of diverse oncogenic alterations.

## MET-driven invasive growth is vulnerable to inhibitors of the integrated stress response

To evaluate the vulnerability of MET-driven invasive growth to ISR inhibition, we analysed the key features of invasive growth with and without ISR inhibitors. We focused on the mRNA levels of genes encoding epithelial markers (CDH1, CDH2, TJP1, and TJP2) and mesenchymal markers (VIM, SNAI1, SNAI2, and ACTA2) in response to HGF stimulation (Dongre and Weinberg, 2019; Zeisberg and Neilson, 2009). As expected, HGF-induced EMT was significantly enhanced under serum starvation, reflecting the increased MET expression under these challenging conditions. Importantly, HGF-driven EMT was effectively inhibited by both GCN2iB and ISRIB (Fig. 5A). Immunofluorescence analysis further supported these findings, showing HGF-induced upregulation of the mesenchymal marker ACTA2 and downregulation of the epithelial marker E-CADHERIN (Fig. 5B). In addition, we examined other MET-induced biological responses under serum

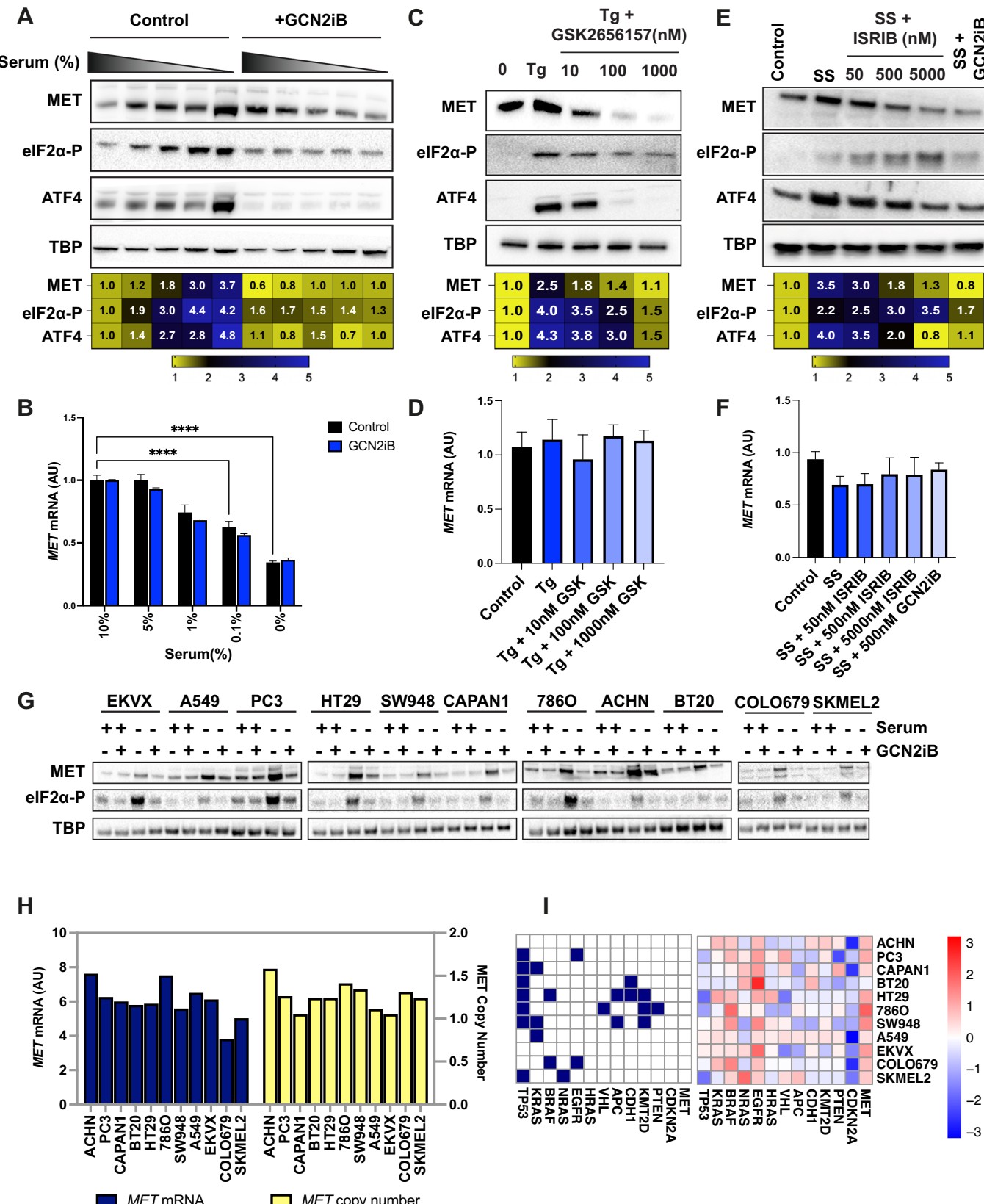

**Figure 4. Stress-induced MET upregulation is impinged by ISR inhibition.**

(A) Western blot analysis of EKVX cells cultured with varying serum concentrations ± 500 nM GCN2iB for 24 h, probing for MET, eIF2α-P, ATF4, and TBP (loading control). Heatmaps depict the relative intensities of bands, where yellow highlights low levels, and blue indicates high levels. (B) Evaluation of *MET* mRNA expression by RT-qPCR in cells subjected to different serum conditions ± GCN2iB, normalised to *TBP* mRNA levels. Error bars represent SEM ($n = 3$ biological replicates). Statistical analysis was performed using two-way ANOVA. (C) Western blot analysis of EKVX cells treated with 500 nM Tg for 6 h. Where indicated, cells were pre-treated with increasing concentrations of the PERK inhibitor GSK2656157 for 1 h before Tg addition. Cells were probed for MET, phosphorylated eIF2α (eIF2α-P), ATF4, and TBP (loading control). The heatmaps below the blots represent the relative band intensities. (D) RT-qPCR analysis of *MET* mRNA levels in EKVX cells treated with 500 nM Tg for 24 h in the presence or absence of the PERK inhibitor GSK2656157 (GSK, 10 nM, 100 nM, or 1000 nM). mRNA levels were normalised to *TBP* mRNA levels, and data are presented as the mean ± SEM ($n = 3$ biological replicates). Statistical analysis was performed using one-way ANOVA with Tukey's post hoc test. (E) The impact of ISR inhibition by ISRIB (0; 50 nM; 500 nM; or 5000 nM) on eIF2α phosphorylation, ATF4, and MET levels under 24 h SS conditions, compared to the control and GCN2iB treatment (500 nM), with TBP as the loading control. (F) RT-qPCR analysis of *MET* mRNA levels in EKVX cells under 24 h serum starvation conditions treated with ISRIB (0, 50 nM, 500 nM, or 5000 nM) or GCN2iB (500 nM). *MET* mRNA levels were normalised to *TBP* mRNA levels, and data are presented as the mean ± SEM. With $n = 3$ biological replicates. Statistical analysis was performed using one-way ANOVA with Tukey's post hoc test. (G) Protein expression analysis via western blot in a panel of cell lines cultured with or without serum and treated with ±GCN2iB (500 nM) for 24 h. (H) Assessment of *MET* mRNA levels and *MET* gene copy number using the CCLE database (Barretina et al, 2012). (I) Mutational status (blue squares indicate mutations) and copy number variations (CNV) in various cancer driver genes based on the CCLE database (Barretina et al, 2012). Significance marker: ****$P < 0.0001$. Source data are available online for this figure.

starvation conditions. Treatment with GCN2iB hampered HGF-induced wound healing (Fig. 5C). ISR inhibitors did not affect cell viability at control conditions, further excluding any unspecific activity and toxicity (Fig. EV5E). Under serum starvation, HGF-induced growth advantage was drastically hindered by ISR inhibition (Fig. 5D). GCN2iB and ISRIB impinged on GEM-induced MET upregulation (Fig. 5E) and hampered MET-fostered resistance to Gemcitabine (Fig. 5F).

Overall, these findings indicate that ISR enhances key features of MET-driven invasive growth.

## Discussion

Our study unveils ISR as a pivotal regulator of *MET* oncogene overexpression in cancer, highlighting a novel layer of translational control that modulates HGF/MET-driven invasive growth.

This discovery extends our understanding beyond the well-documented transcriptional regulation of the *MET* oncogene, shedding light on a novel, post-transcriptional mechanism that cancer cells harness to escalate *MET* expression in response to environmental stress. The complexity of cellular protein expression regulation is well-established, encompassing both transcriptional control mechanisms and translational tuning (Ghaemmaghami et al, 2003; Li et al, 2014; Wullschleger et al, 2006). Specifically, the translational tuning uses the pool of already available mRNA. It is a critical mechanism for promptly regulating protein synthesis, prioritising the cellular response to environmental stimuli (Pakos-Zebrucka et al, 2016). The *MET* oncogene exquisitely exemplifies these multiple layers of regulation, with our previous work highlighting the inducible nature of the *MET* promoter (Gambarotta et al, 1994) and how *MET* expression is transcriptionally modulated in response to hypoxia or genetic damage caused by irradiation (De Bacco et al, 2011; Pennacchietti et al, 2003).

Our findings reveal the translational control exerted by the 5'UTR of *MET* mRNA as a critical regulatory layer, representing a sophisticated cellular strategy for prompt adaptation. Under normal conditions, *MET* is expressed at low levels, primed for swift activation to facilitate immediate responses like cell scattering or migration, which are crucial for tissue repair upon external cues (Matsumoto et al, 2014; Trusolino et al, 2010). The ubiquity of the MET ligand, HGF, further underscores the system readiness

for activation, with HGF presence in the extracellular matrix ensuring that MET signalling can be promptly engaged (Matsumoto et al, 2014; Nakamura et al, 1989; Naldini et al, 1995). Thus, a response that enables swift MET synthesis is essential for general homoeostasis and survival. These mechanisms are primarily prone to be usurped by cancer cells that evolve in a hostile milieu (Anderson and Simon, 2020). Therefore, understanding the molecular mechanisms that foster *MET* expression is pivotal.

The discovery of upstream open reading frames (uORFs) by Marilyn Kozak in the early 1980s laid the groundwork for our current exploration of *MET* regulation (Kozak, 1987). Unlike ATF4, known for its regulation through the ISR and serving as a paradigm of translational unleash under stress (Foiani et al, 1991; Lu et al, 2004; Vattem and Wek, 2004), MET involvement in similar regulatory pathways remained unexplored until now. Here, we show that *MET* 5'UTR harbours two uORFs, elements that are key to its regulation. These uORFs are pivotal for translational regulation under stress, mirroring the established mechanism governing the translation of *ATF4*. ISR modulates ATF4 synthesis via eIF2α phosphorylation, a process that similarly governs *MET* translation by circumventing the inhibitory effects of uORFs (Costa-Mattioli and Walter, 2020; Pakos-Zebrucka et al, 2016). Consequently, our findings elevate MET to a novel stress marker in cancer.

Far from being a passive marker of ISR activation, our findings underscore MET's active participation in the cellular response to environmental challenges. We demonstrate a modest yet consistent three- to fivefold increase in MET protein levels in response to many stress stimuli relevant to the adverse tumour microenvironment. This upregulation is not merely a biochemical footnote but has significant implications for cancer cells. Specifically, we show that mutations within the *MET* mRNA uORFs, which render MET insensitive to stress signals, or the pharmacological inhibition of the ISR, significantly impair epithelial-to-mesenchymal transition, wound healing, and cell survival.

Moreover, while the linkage between the ISR pathway and cancer development is increasingly recognised (Costa-Mattioli and Walter, 2020; Gavish et al, 2023), our study highlights the interplay between ISR activation and oncogenic signalling pathways, a domain that remains partially charted. By underscoring the ISR-induced post-transcriptional modulation of *MET*, we unveil a

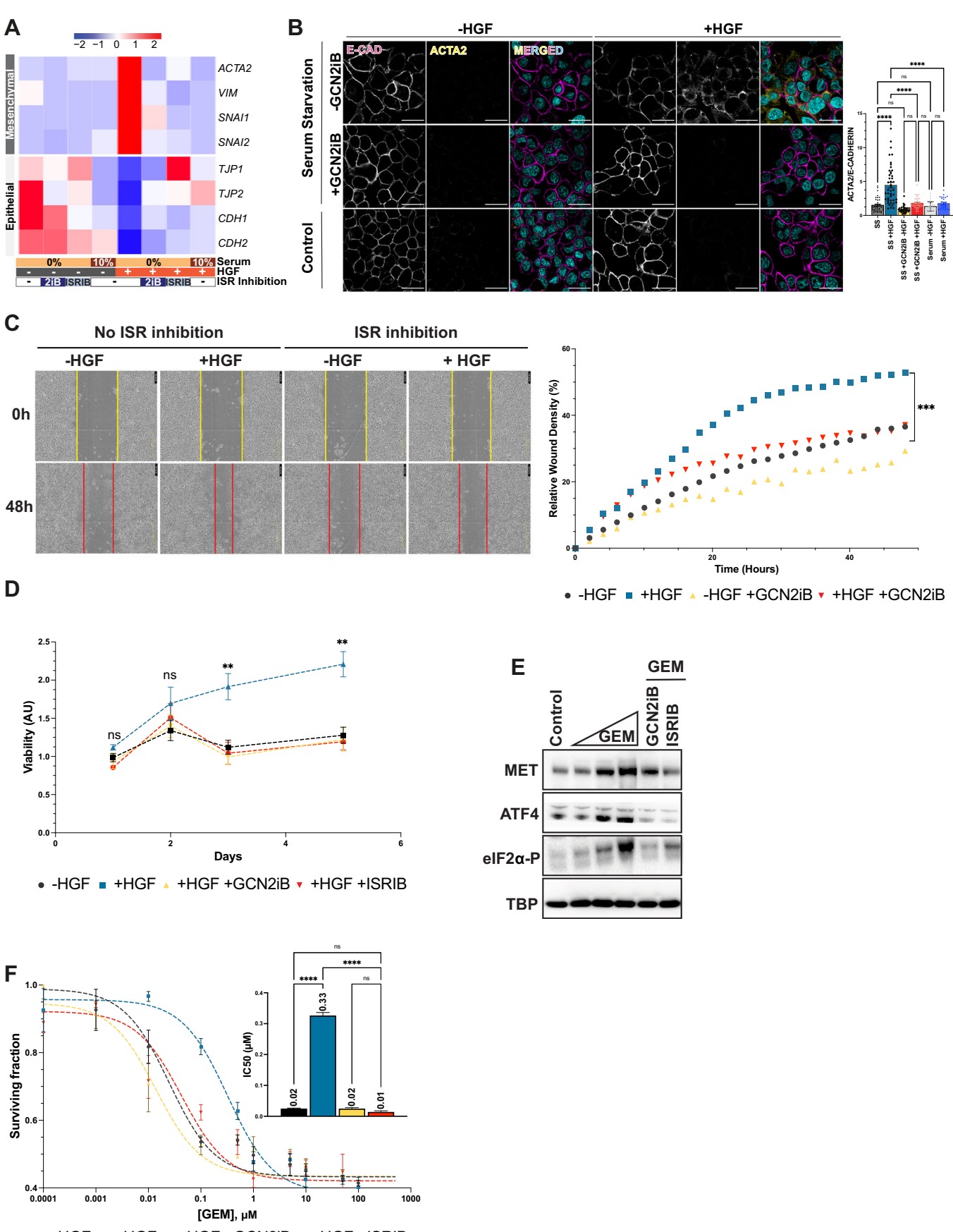

**Figure 5. MET-driven invasive growth is vulnerable to inhibitors of the integrated stress response.**

(A) Heatmap visualisation of EMT marker expression profiles obtained by RT-qPCR in EKVX cells cultured under various conditions for 72 h. Red indicates upregulation, and blue indicates downregulation of gene expression normalised to *TBP* expression. (B) Immunofluorescence staining of EKVX cells for E-Cadherin (epithelial marker, magenta) and ACTA2 (mesenchymal marker, yellow) under control (10% serum) or SS for 72 h ± 500 nM GCN2iB ± 50 ng/mL HGF. Nuclei counterstained with DAPI (cyan). Scale bar: 25 µm. Fluorescence intensity was quantified using ImageJ software from $n > 3$ biological replicates. Two-way ANOVA with repeated measures was used for statistical analysis. Represented $P$ values: SS vs. SS + HGF: $4.11 \times 10^{-4}$ (****); SS + HGF vs. SS + HGF + GCN2iB: $6.03 \times 10^{-4}$ (****); SS - HGF + GCN2iB vs. SS + HGF + GCN2iB: 0.054 ("ns"); SS vs. SS - HGF + GCN2iB: 0.25 ("ns"); + serum - HGF vs. + serum + HGF: 0.065 ("ns"); SS + HGF vs. + serum + HGF: $2.1 \times 10^{-5}$ (****); SS - HGF vs. + serum - HGF: 0.856 ("ns"). (C) Kinetic analysis of wound healing in EKVX cells under serum starvation conditions ± 500 nM GCN2iB ± 50 ng/mL HGF. Wound density was monitored every 2 h. Data presented as relative wound density (%) are mean ± SEM, with $n = 6$ biological replicates. Results were analysed with two-way ANOVA with repeated measures. Represented p values for control vs. +HGF comparisons at the end of the experiment: $P = 2.22 \times 10^{-4}$ (***). (D) Viability of EKVX cells measured over time under serum starvation ± 50 ng/ml HGF ± 500 nM GCN2iB ± 1 µM ISRIB. Results are normalised to initial viability (day 1) and are represented as mean ± SEM with $n = 6$ biological replicates. Data were analysed using two-way ANOVA with repeated measures. Represented $P$ values for control vs. +HGF comparisons: day 1: 0.2493 (ns); day 2: 0.1302 (ns); day 3: $3.5 \times 10^{-3}$ (**); day 4: $2.2 \times 10^{-3}$ (**). Comparisons of +HGF + ISRIB vs. control and +HGF + GCN2iB vs. control remained "ns" throughout the experiment. (E) Western blot analysis of MET, ATF4, and eIF2α-P levels in EKVX cells after treatment with increasing GEM concentrations (5 nM; 50 nM; 500 nM). For ISR inhibition (with either 500 nM GCN2iB or 1 µM ISRIB), cells were treated with 500 nM of GEM. TBP is used as a loading control. (F) Dose–response survival curves of EKVX cells treated with GEM for 48 h across a range of concentrations, with or without additional treatments (50 ng/mL HGF, 500 nM GCN2iB, or 1 µM ISRIB). The inset shows $IC_{50}$ quantifications for each treatment condition. Results are presented as mean ± SEM ($n > 3$ biological replicates) and analysed using one-way ANOVA with Tukey's post hoc test. Significance markers: ns (not significant), $**P < 0.01$, $***P < 0.001$, $****P < 0.0001$. Source data are available online for this figure.

detailed portrait of how ISR integrates into the MET-driven invasive growth programme of cancer cells.

It is important to note that the biological features driven by the ISR/MET axis were measured in several cancer cells with variegated genetic landscapes where MET is unlikely to serve as a primary driver gene. Nonetheless, it is well-established that, in many instances, MET is responsible for promoting cancer progression, rather than initiation. It is known that genetic alterations of some driver oncogenes activate ISR, ultimately leading to MET over-expression (Denoyelle et al, 2006; Hart et al, 2012). This concept highlights both the widespread MET overexpression in cancer and its role in cancer progression.

In a clinical perspective, activation of the Integrated Stress Response (ISR) by chemotherapeutic agents or ionising radiations, and the resulting MET upregulation, indicate that MET can be actively involved in supporting mechanisms of adaptive resistance to therapy. In the ISR context, MET targeting as a strategy to enhance therapeutic efficacy and prevent resistance to standard treatments deserves further investigation.

# Methods

**Reagents and tools table**

| Reagent/resource | Reference or source | Identifier or catalogue number |
|---|---|---|
| **Experimental models** | | |
| HEK-293 (*H. sapiens*) | ATCC | RRID:CVCL_0045 |
| EKVX (*H. sapiens*) | NCI | RRID:CVCL_1195 |
| A549 (*H. sapiens*) | ATCC | RRID:CVCL_0023 |
| PC-3 (*H. sapiens*) | ATCC | RRID:CVCL_0035 |
| HT-29 (*H. sapiens*) | ATCC | RRID:CVCL_0320 |
| SW948 (*H. sapiens*) | ATCC | RRID:CVCL_0632 |
| CAPAN-1 (*H. sapiens*) | ATCC | RRID:CVCL_0237 |
| 786-O (*H. sapiens*) | ATCC | RRID:CVCL_1051 |
| ACHN (*H. sapiens*) | ATCC | RRID:CVCL_1067 |

| Reagent/resource | Reference or source | Identifier or catalogue number |
|---|---|---|
| BT-20 (*H. sapiens*) | ATCC | RRID:CVCL_0178 |
| COLO-679 (*H. sapiens*) | DSMZ | RRID:CVCL_1130 |
| SK-MEL-2 (*H. sapiens*) | ATCC | RRID:CVCL_0069 |
| **Recombinant DNA** | | |
| pGL4.23[luc2/minP] | Promega | E8411 |
| pGL4.75[hRLuc/CMV] | Promega | E6911 |
| pGL4.23[METUTRwt/ luc2/minP] | Addgene | 226458 |
| pGL4.23[uORF1/luc2/ minP] | Addgene | 226459 |
| pGL4.23[uORF2/luc2/ minP] | Addgene | 226460 |
| pGL4.23[uORF1,2/luc2/ minP] | Addgene | 226461 |
| pLenti-IRES-GFP | Addgene | 169280 |
| pLenti-Exon1-MET | Addgene | 216535 |
| pLenti-Exon2-MET | Addgene | 216536 |
| pLenti_5′UTR_MET_GFP | Addgene | 214103 |
| pLenti-uORF1-MET | Addgene | 214104 |
| pLenti-uORF2-MET | Addgene | 214105 |
| pLenti-uORF1,2-MET | Addgene | 214106 |
| Rluc_IRES_Fluc | Addgene | 45642 |
| Rluc_IRES_Fluc_Pless | Addgene | 226462 |
| Rluc_MET_95_Fluc | Addgene | 226463 |
| Rluc_MET_95_Fluc_Pless | Addgene | 226464 |
| pEF1a-hMLH1dn | Addgene | 174824 |
| pCMV-PEmax_p2A_GFP | Addgene | 180020 |
| pU6-pegRNA-GG-acceptor | Addgene | 132777 |
| pU6-sgRNA_MET_uORF1 | Addgene | 216537 |

| Reagent/resource | Reference or source | Identifier or catalogue number |
| --- | --- | --- |
| pU6-sgRNA_MET_uORF2 | Addgene | 216538 |
| **Antibodies** | | |
| E-CADHERIN | Abcam | ab40772 |
| ACTA2 | Sigma Aldrich | A2547 |
| P-MET-Y1234/1235 | Cell Signalling Technology | 3077 |
| Total MET | Cell Signalling Technology | 3148 |
| P-eIF2α-S52 | Cell Signalling Technology | 3398 |
| ATF4 | Cell Signalling Technology | 11815 |
| Total eIF2α | Cell Signalling Technology | 5324 |
| P-AKT-S473 | Cell Signalling Technology | 4060 |
| Total AKT | Cell Signalling Technology | 9272 |
| TBP | Abcam | ab220788 |
| **Oligonucleotides and other sequence-based reagents** | | |
| TJP1 | ThermoFisher Scientific | Hs01551861_m1 |
| TJP2 | ThermoFisher Scientific | hs00910543_m1 |
| CDH1 | ThermoFisher Scientific | hs00170423_m1 |
| CDH2 | ThermoFisher Scientific | hs00169953_m1 |
| ACTA2 | ThermoFisher Scientific | hs00426835_g1 |
| VIM | ThermoFisher Scientific | hs00185584_m1 |
| SNAI1 | ThermoFisher Scientific | hs00195591_m1 |
| SNAI2 | ThermoFisher Scientific | hs00950344_m1 |
| ATF4 | ThermoFisher Scientific | Hs00909569_g1 |
| MET | ThermoFisher Scientific | hs01565584_m1 |
| DDIT3 | ThermoFisher Scientific | Hs00358796_g1 |
| ATF3 | ThermoFisher Scientific | Hs00231069_m1 |
| GADD45A | ThermoFisher Scientific | hs00169255_m1 |
| TBP | ThermoFisher Scientific | Hs00427620_m1 |
| gRNA targeting MET exon 6 | Designed in study | 5'-GAGAGCACGAU GAAUACGUA-3' |
| gRNA targeting eIF2α S52A - gRNA1 | Designed in study | 5'-CUGGAAUACAA CAACAUUGA-3' |

| Reagent/resource | Reference or source | Identifier or catalogue number |
| --- | --- | --- |
| gRNA targeting eIF2α S52A - gRNA2 | Designed in study | 5'-GAUAGAACGGA UACGCCUUC-3' |
| gRNA targeting eIF2α S52A - gRNA3 | Designed in study | 5'-UUCUUAGUGA AUUAUCCAGA-3' |
| Donor DNA 1 eIF2α S52A | Designed in study | 5'-GGCTTATGTCAGCTT GCTGGAATACAACAA CATCGAAGGCATGATTCTT CTTG CCGAATTAGCCAGAA GGCGTATCCGTT CTATCAACAA ACTCATCCGA-3' |
| Donor DNA 2 eIF2α S52A | Designed in study | 5'-ACAACATTGAAG GCATGATTCTTCTT GGTGAATTAGCC AGGAGGCGTATCC G TTCTATCAACAAA CTCATCCGAATT-3' |
| Donor DNA 3 eIF2α S52A | Designed in study | 5'-ACAACATTGAA GGCATGATTC TTCTTgGTGAATTA GCCAGAAGGCGT ATCCG TTCTATCAA CAAACTCATCCG-3' |
| **Chemicals, enzymes and other reagents** | | |
| Lipofectamine 3000 reagent | ThermoFisher Scientific | L3000001 |
| DMEM-LM | ThermoFisher Scientific | 30030 |
| L-Glutamine | ThermoFisher Scientific | A2916801 |
| L-Methionine | Sigma Aldrich | M5308 |
| L-Leucine | Sigma Aldrich | L8912 |
| GCN2iB | Selleckchem | S8929 |
| GSK2656157 | Selleckchem | S7033 |
| ISRIB | Selleckchem | S0706 |
| Salubrinal | Selleckchem | S2923 |
| Thapsigargin | Selleckchem | S7895 |
| GEM | Selleckchem | S1714 |
| Recombinant human HGF protein | R&D Systems | 294-HG/CF |
| GeneArt Genomic Cleavage Detection Kit | ThermoFisher Scientific | A24372 |
| Lipofectamine CRISPRMAX Cas9 Transfection Reagent | ThermoFisher Scientific | CMAX00008 |
| TrueCut Cas9 Protein v2 | ThermoFisher Scientific | A36498 |
| Invitrolon PVDF membranes | ThermoFisher Scientific | LC2005 |
| ECL Prime Western Blotting Detection Reagent | Merck | GERPN2232 |
| ECL Select Western Blotting Detection Reagent | Merck | GERPN2235 |

| Reagent/resource | Reference or source | Identifier or catalogue number |
|---|---|---|
| Promega Luciferase Assay System | Promega | E1500 |
| CyQUANT Direct Cell Proliferation Assay | ThermoFisher Scientific | C34011 |
| Incucyte® Wound Maker 96-Tool | Incucyte | 4563 |
| VECTASHIELD® Antifade Mounting Medium | Vector Laboratories | H-1200-10 |
| Maxwell RSC miRNA Tissue Kit | Promega | AS1460 |
| SuperScript III First-Strand Synthesis System | ThermoFisher Scientific | 18080-051 |
| **Software** | | |
| GraphPad Prism 10.2.3 | https://www.graphpad.com | |
| SnapGene 7.2.1 | https://www.snapgene.com/ | |
| ImageJ | https://imagej.nih.gov/ij/index.html | |

## Plasmids

All plasmids used in the study are listed in the Reagents and Tools table and are available from Addgene.

## Cell culture and transfection

Cell lines employed in this study are detailed in the Reagents and Tools table. All cell lines were maintained at 37 °C in a humidified atmosphere with 5% $CO_2$. The IFOM Cell Culture Service performed routine mycoplasma testing, confirming our cultures were consistently contamination-free. Lipofectamine 3000 reagent (Catalogue number: L3000001, ThermoFisher Scientific) was utilised according to the manufacturer's recommendations for transfection protocols.

## Stress inductions and treatments

All cell lines were plated in RPMI1640 + 2mM L-Glutamine (Catalogue number: A2916801, ThermoFisher Scientific) + 10% foetal bovine serum (FBS, complete medium) at 70% confluency. Cells were washed twice with RPMI1640 medium without FBS and without L-Glutamine. For serum starvation, cells were maintained in serum-deprived RPMI1640 for the indicated time. For hypoxia experiments, cells were maintained in complete medium in 3% $O_2$ and harvested at indicated times. For ionising radiation, cells were irradiated with 10 Gy (2.5 Gy/min) using the Faxitron X-Ray System (Model number 43855F, British Medical Auctions) and incubated for recovery for the indicated times and processed for western blot. For GEM (Catalogue number S1714, Selleckchem) treatments, cells were grown in complete medium with indicated concentrations of GEM for 24–48 h. ER stress was induced using thapsigargin (Catalogue number S7895, Selleckchem). Leucine deprivation was performed as previously described (Harding et al, 2000; Harding et al, 2003).

For ISR inhibition, cells were treated with GCN2iB (Catalogue number S8929, Selleckchem), GSK2656157 (Catalogue number S7033, Selleckchem), or ISRIB (Catalogue number S0706, Selleckchem) for the whole duration of the experiments. For sustained eIF2α phosphorylation, cells were treated with salubrinal (Catalogue number S2923, Selleckchem).

When indicated, cells were treated with 50 ng/ml of carrier-free recombinant human HGF protein (Catalogue number 294-HG/CF, R&D Systems) for 15 min.

## CRISPR/Cas9 genome editing

### MET knockout in EKVX cells
The genome editing of EKVX cells to knock out the *MET* gene was executed by Cogentech Genome Editing Service. A guide RNA (gRNA) specifically designed to target exon 6 of the *MET* gene (sequence: 5'-GAGAGCACGAUGAAUACGUA-3') was used. Successful editing was confirmed using PCR, Sanger sequencing (Forward: 5'-TGTGGGAAAATGAAAGAATTTCAGA-3'; Reverse: 5'-CGTGGGGATGGTTCAGTGAG-3'), and western blotting to assess the absence of MET expression.

### eIF2α S52A mutation
For the introduction of the S52A mutation into the *eIF2α* gene, three guide gRNAs were designed to target the region of interest. Sequences are available in the Reagents and Tools table.

The editing efficiency of each gRNA was verified by GeneArt™ Genomic Cleavage Detection Kit according to the manufacturer's instructions (Catalogue number: A24372, ThermoFisher Scientific). Transfections were performed using the Lipofectamine™ CRISPR-MAX™ Cas9 Transfection Reagent (Catalogue number: CMAX00008, ThermoFisher Scientific) using TrueCut™ Cas9 Protein v2 (Catalogue number: A36498, ThermoFisher Scientific), following the manufacturer's protocol. gRNAs were used alongside the following donor DNAs to introduce the S52A mutation through Homology Directed Repair (Reagents and Tools table).

Monoclonal cells were obtained by limiting dilution by IFOM Cell Culture Service. The genomic DNA was extracted from the cells, and the eIF2alpha gene was amplified using PCR with primers (Forward: 5'-CTTTCAGTGGCAGGATGTGG-3'; Reverse: 5'-CACACACT-CATTCCTGCCAA-3'). The amplified products were then subjected to Sanger sequencing to confirm the presence of the S52A mutation. Similar levels of total eIF2α to parental cells and the absence of stress-induced phosphorylation were confirmed by western blot.

### MET upstream open reading frame (uORF) mutations
The mutations in the *MET* gene's upstream open reading frame (uORF) were generated using the Prime Editing (Anzalone et al, 2019; Chen et al, 2021). The specific plasmids used for this purpose are detailed in the Reagents and Tools table.

## Western blotting

The cells were lysed as previously described (Cerqua et al, 2022) using Laemmli Lysis Buffer. Proteins were transferred onto Invitrolon™ PVDF membranes (Catalogue number LC2005, ThermoFisher Scientific) and probed with antibodies specified in the Reagents and Tools table. A chemiluminescent reaction was used to visualise the protein bands, employing either ECL™ Prime Western Blotting Detection Reagent

(Catalogue number GERPN2232, Merck) or ECL™ Select Western Blotting Detection Reagent (Catalogue number GERPN2235, Merck).

## Luciferase reporter assay

Luciferase assays were performed using the Promega Luciferase Assay System (Catalogue number E1500, Promega) following the manufacturer's protocol. Briefly, cells were plated in 96-well plates at a density of 10,000 cells per well and transfected with the luciferase reporter plasmid. After 24 h, indicated treatments were applied. Cells were lysed 24 h post-ISR induction, and luciferase activity was measured using a luminometer and the Promega Luciferase Assay System.

## Lentiviral packaging

Lentiviral packaging was performed using standard molecular biology techniques. HEK293T cells were plated in 10-cm dishes at a $5 \times 10^6$ cells/dish density in DMEM medium supplemented with 10% FBS + 2 mM L-Glutamine. The next day, cells were transfected with 5 µg of lentiviral expression vector, 4 µg of psPAX2 packaging vector, and 1 µg of pMD2.G envelope vector. After 6 h, the transfection medium was replaced with complete fresh medium. The lentivirus-containing supernatant was collected 48 h post-transfection and filtered through a 0.45-µm pore size filter to remove cell debris. The lentiviral titre was determined by flow cytometry using the GFP reporter, and the lentivirus was stored at −80 °C until use.

## Immunofluorescence

Immunofluorescence experiments were performed as described in (Altintas et al, 2022). Briefly, cells were fixed with 4% paraformaldehyde, permeabilised with 0.1% Triton X-100 and blocked with 1% bovine serum albumin. The cells were then incubated with primary antibodies (Reagents and Tools table) overnight at 4 °C, followed by incubation with fluorescent secondary antibodies for 1 h at room temperature. The coverslips were mounted on glass slides using a mounting medium containing DAPI (VECTASHIELD® Antifade Mounting Medium, catalogue number H-1200-10, Vector Laboratories).

## Reverse transcription-quantitative polymerase chain reaction

Total RNA was isolated from cells using the Maxwell RSC miRNA Tissue Kit (Catalogue number AS1460, Promega) with the Maxwell RSC Instrument and reverse transcribed using the SuperScript III First-Strand Synthesis System (ThermoFisher Scientific™, Catalogue number 18080-051, ThermoFisher Scientific). qRT-PCR was performed using the TaqMan™ probes detailed in the Reagents and Tools table.

## Viability assay

Cells were plated in 96-well plates at a density of 5000 cells per well. Cells were then treated with various compounds or conditions, as indicated in the Figure legends. Following treatments, the cells were incubated with the CyQUANT™ Direct Cell Proliferation Assay (ThermoFisher Scientific, Catalogue number C34011) reagent as per the manufacturer's instructions.

## Wound-healing assay

EKVX cells were plated on 96-well plates in complete medium and let adhere overnight. The scratch was performed with Incucyte® Wound Maker 96-Tool (Catalogue number 4563), and cells were washed to remove debris, maintained in serum-free medium ± indicated treatments for the duration of the experiments. The wound area was monitored using The Incucyte® Scratch Wound Analysis Software Module as Relative wound density, which quantifies the cell density in the wound area relative to the cell density outside the wound every 2 h.

## Statistical analysis

All experiments were performed at least in triplicates and repeated at least three times. Statistical analyses were performed using appropriate tests as indicated in the Figure legends. All statistical analyses were conducted using GraphPad Prism software (GraphPad Software Inc.). The data are presented as mean ± SEM. Statistical significance was determined using a two-tailed Student's $t$ test for comparing two groups or ANOVA with Tukey's post hoc test for comparing multiple groups. In cases of repeated measurements (e.g., time-course experiments), a mixed-effects model was used to account for both fixed and random effects, with $P$ values calculated for time, treatment conditions, and their interaction. $P < 0.05$ was considered statistically significant.

# Data availability

Further information and requests for data and reagents should be directed to and will be fulfilled by the corresponding author, Dogus M Altintas (dogus.altintas@ifom.eu) without restriction. This study includes no data deposited in external repositories.

The source data of this paper are collected in the following database record: biostudies:S-SCDT-10_1038-S44318-024-00338-4.

# Peer review information

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

## Acknowledgements

We want to IFOM Imaging Unit, Cogentech Genome Editing Service, Cogentech DNA Sequencing Service, and Cogentech Quantitative PCR Service for their technical assistance. We particularly acknowledge IFOM Cell Biology Unit, especially Ilaria Rancati, Alicia Rubio and Stefania Lavore for the critical assistance in CRISPR/Cas9-edtied cells' selection. All illustrations were Created with BioRender.com. This work was supported by the Associazione Italiana per la Ricerca sul Cancro (AIRC), N. 23820 to PMC and N. 28836 to CB, and the Fondazione Umberto Veronesi post-doctoral fellowship to DMA (Grant number 4691).

## Author contributions

**Marina Cerqua**: Investigation; Methodology. **Marco Foiani**: Conceptualisation. **Carla Boccaccio**: Funding acquisition; Writing—original draft; Writing—review and editing. **Paolo M Comoglio**: Conceptualisation; Supervision; Funding acquisition; Visualisation; Writing—original draft; Writing—review and editing. **Dogus M Altintas**: Conceptualisation; Formal analysis; Supervision; Funding acquisition; Validation; Investigation; Visualisation; Methodology; Writing—original draft; Writing—review and editing.

Source data underlying figure panels in this paper may have individual authorship assigned. Where available, figure panel/source data authorship is listed in the following database record: biostudies:S-SCDT-10_1038-S44318-024-00338-4.

## Disclosure and competing interests statement

The authors declare no competing interests.

# Expanded View Figures

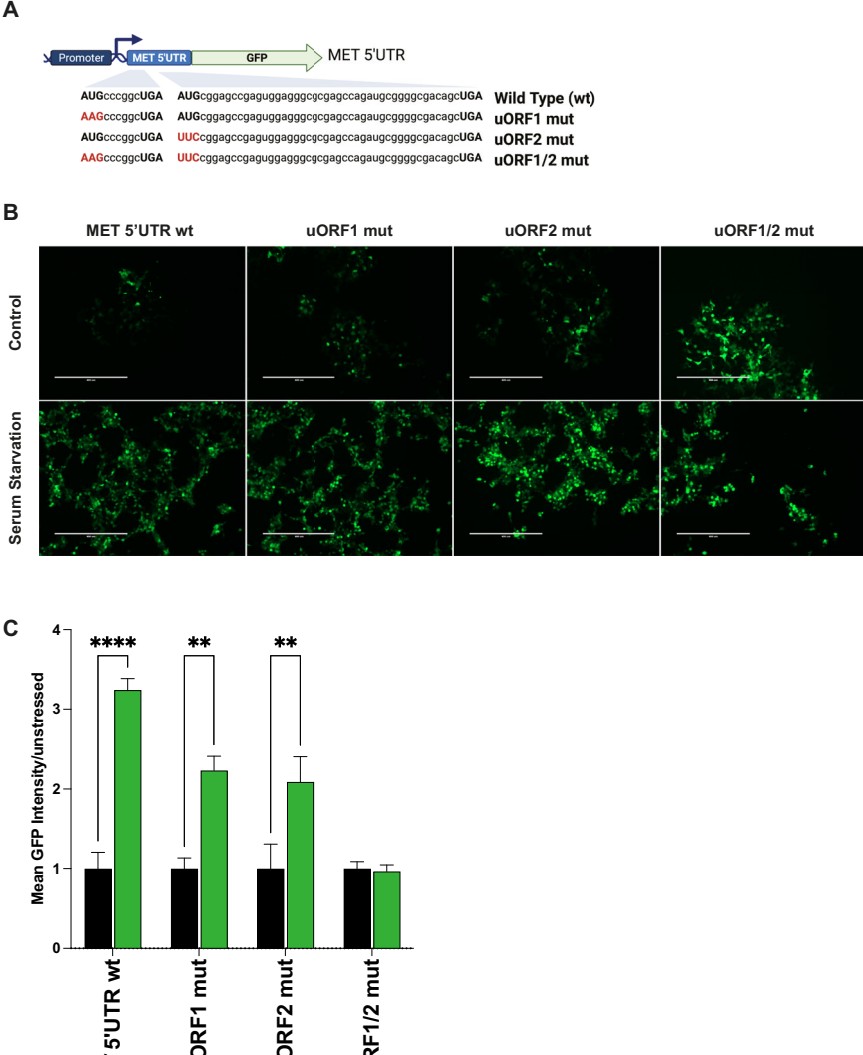

**Figure EV1.**  ***MET* mRNA translation is upregulated by a broad range of stress stimuli through the 5'UTR.**

(**A**) Schematic representation of the GFP reporter construct under the control of the MET 5'UTR. Sequences depict wild-type (wt) and mutant (mut) versions of uORF1, uORF2, or the double mutant uORF1/2. (**B**) Fluorescence microscopy images of cells transfected with constructs containing the MET wt 5'UTR or its uORF mutants cultured under control conditions or serum starvation. Scale bars represent 400 μm. (**C**) Quantification of GFP fluorescence intensity from images in (**B**). Data are presented as mean GFP intensity normalised to the control condition for each construct. Results are mean ± SEM ($n = 3$ biological replicates). The *P* value is calculated using one-way ANOVA with Tukey's post hoc test. Represented *P* values for control vs. serum starvation wt: $4.47 \times 10^{-5}$ (****); control vs. serum starvation uORF1 mut: $2.01 \times 10^{-3}$ (**); control vs. serum starvation uORF2 mut: $5.80 \times 10^{-3}$ (**); control vs. serum starvation uORF1/2 mut: 0.995 (non-significant).

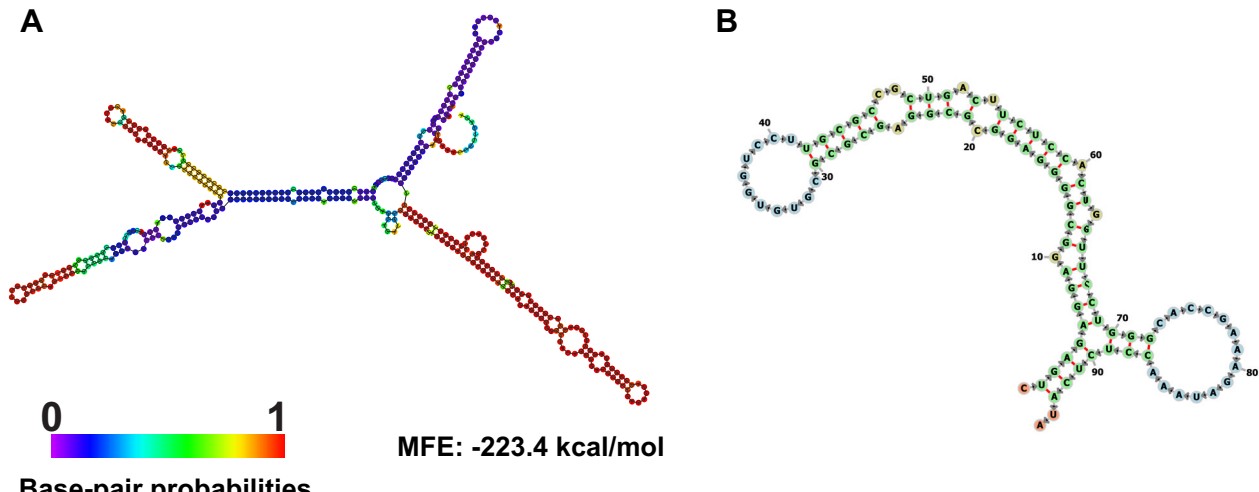

**A**

0 — 1
**Base-pair probabilities**

**MFE: -223.4 kcal/mol**

**B**

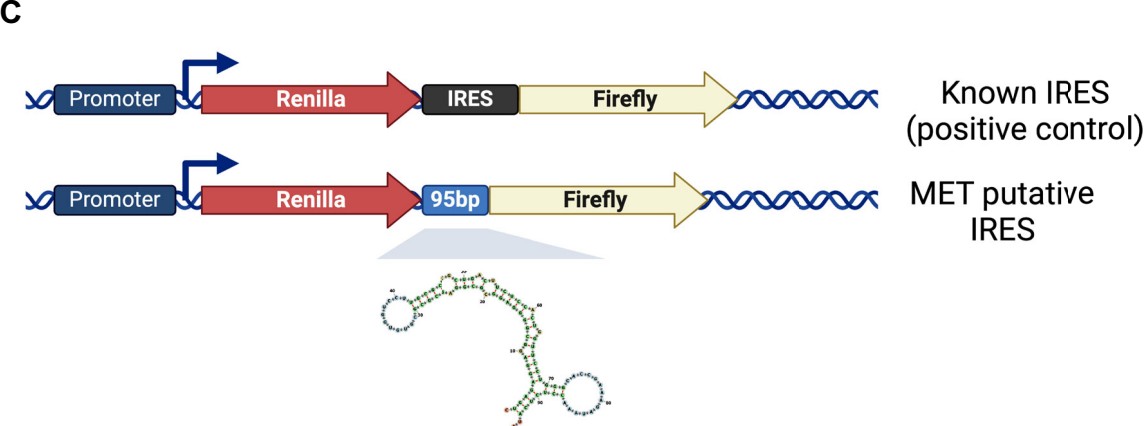

**C**

Known IRES (positive control)

MET putative IRES

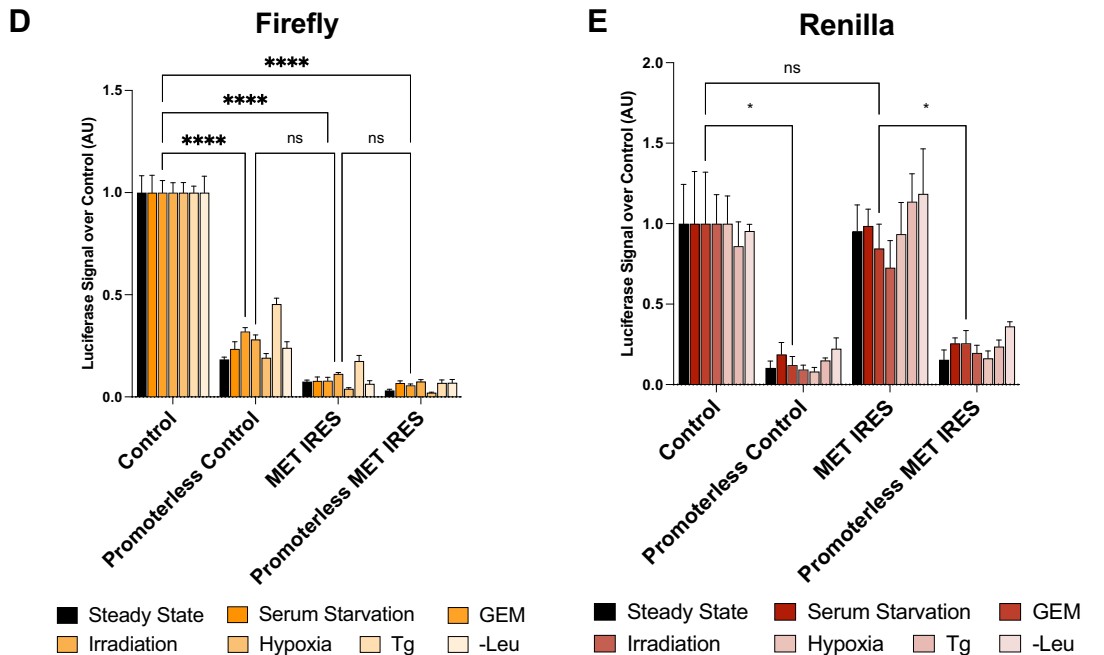

**D** Firefly

**E** Renilla

**Figure EV2.  Structural prediction of *MET* 5′UTR and functional assessment of a putative *MET* IRES.**

(A) Predicted secondary structure of the *MET* 5′UTR as determined by RNAfold (Lorenz et al, 2011). The structure is colour-coded according to base-pair probabilities, with the minimum free energy (MFE) calculated to be −223.4 kcal/mol. (B) Secondary structure of the putative *MET* IRES as described by the human IRES atlas (Yang et al, 2021). The IRES is 95 bases in length, starting at base 302 and ending at base 396. (C) Schematic representation of the bicistronic reporter constructs used to assess IRES activity. The predicted *MET* IRES sequence was cloned between the Renilla luciferase (RLuc) and Firefly luciferase (FLuc) genes. A vector containing a known functional IRES sequence was used as a positive control (Poulin et al, 1998). (D) Firefly Luciferase measurements assessing the activity of the MET putative IRES in comparison to a known IRES sequence. Promoterless vectors were used as negative controls. The error bars represent mean ± SEM with n = 3 biological replicates. The $P$ value is calculated using two-way ANOVA with Tukey's post hoc test and represents the row factor (different constructs) and column factor (different treatments). (E) Renilla results reflect promoter-driven expression, showing that the CMV promoter drives the first cistron consistently across conditions, ensuring that the system functions as expected. Results are represented and analysed as described in (D). Significance markers: ****$P < 0.0001$, *$P < 0.05$, and "ns" indicates not significant.

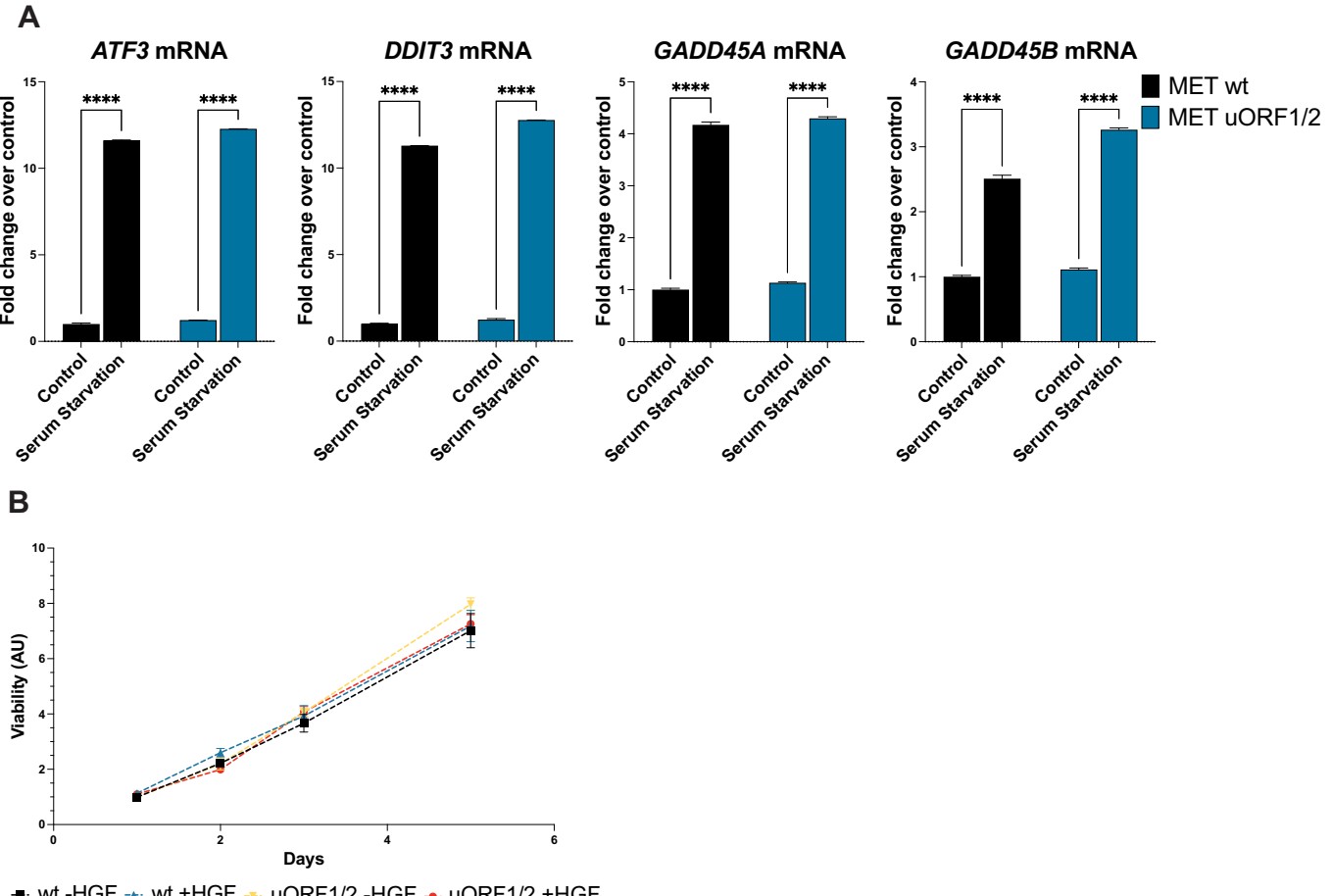

**A**

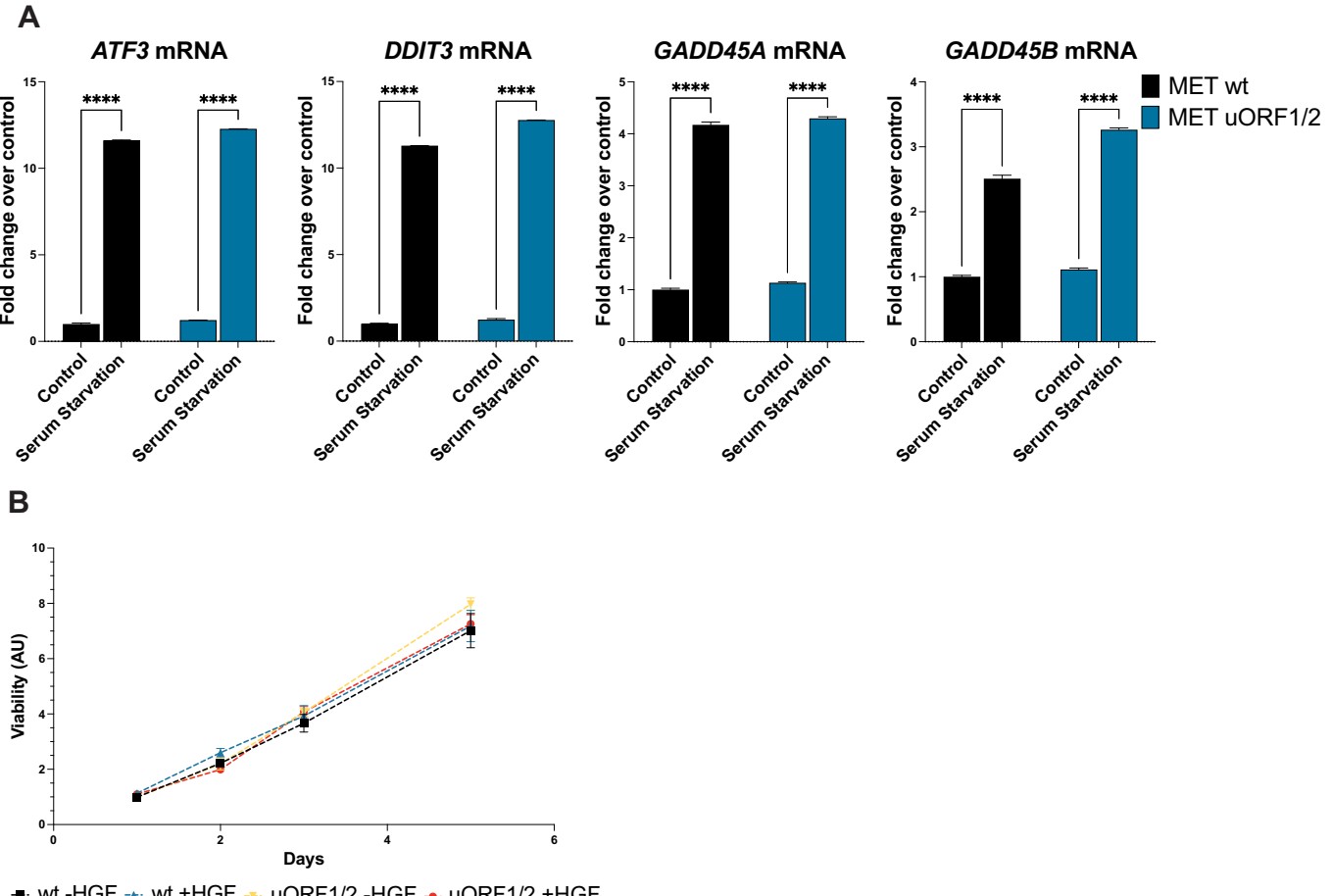

**B**

**Figure EV3.** *MET translation and the subsequent biological activity are regulated by the two uORFs.*

(A) Bar graphs presenting the relative mRNA levels of *ATF3, DDIT3, GADD45A,* and *GADD45B* in cells expressing either MET wt or MET 5′UTR with uORF1/2 mutations. The cells were subjected to ± serum starvation. Data are normalised to *TBP* expression levels. Results are represented as mean ± SEM ($n = 3$ biological replicates) and analysed by one-way ANOVA with Tukey post hoc test. Exact *P* values: ATF3 (Control vs. Serum Starvation, MET wt: $P = 4.56 \times 10^{-6}$, MET uORF1/2: $P = 1.67 \times 10^{-5}$); DDIT3 (Control vs. Serum Starvation, MET wt: $P = 2.09 \times 10^{-5}$, MET uORF1/2: $P = 3.98 \times 10^{-5}$); GADD45A (Control vs. Serum Starvation, MET wt: $P = 2.11 \times 10^{-5}$, MET uORF1/2: $P = 1.12 \times 10^{-5}$); GADD45B (Control vs. Serum Starvation, MET wt: $P = 1.33 \times 10^{-5}$, MET uORF1/2: $P = 9.08 \times 10^{-5}$). (B) Viability assay showing the growth curves of cells over six days under control conditions, comparing wild-type with uORF1/2 double-mutant cells ± HGF. Two-way ANOVA with repeated measures was used to analyse data ($n > 3$ biological replicates). Significance marker: ****$P < 0.0001$.

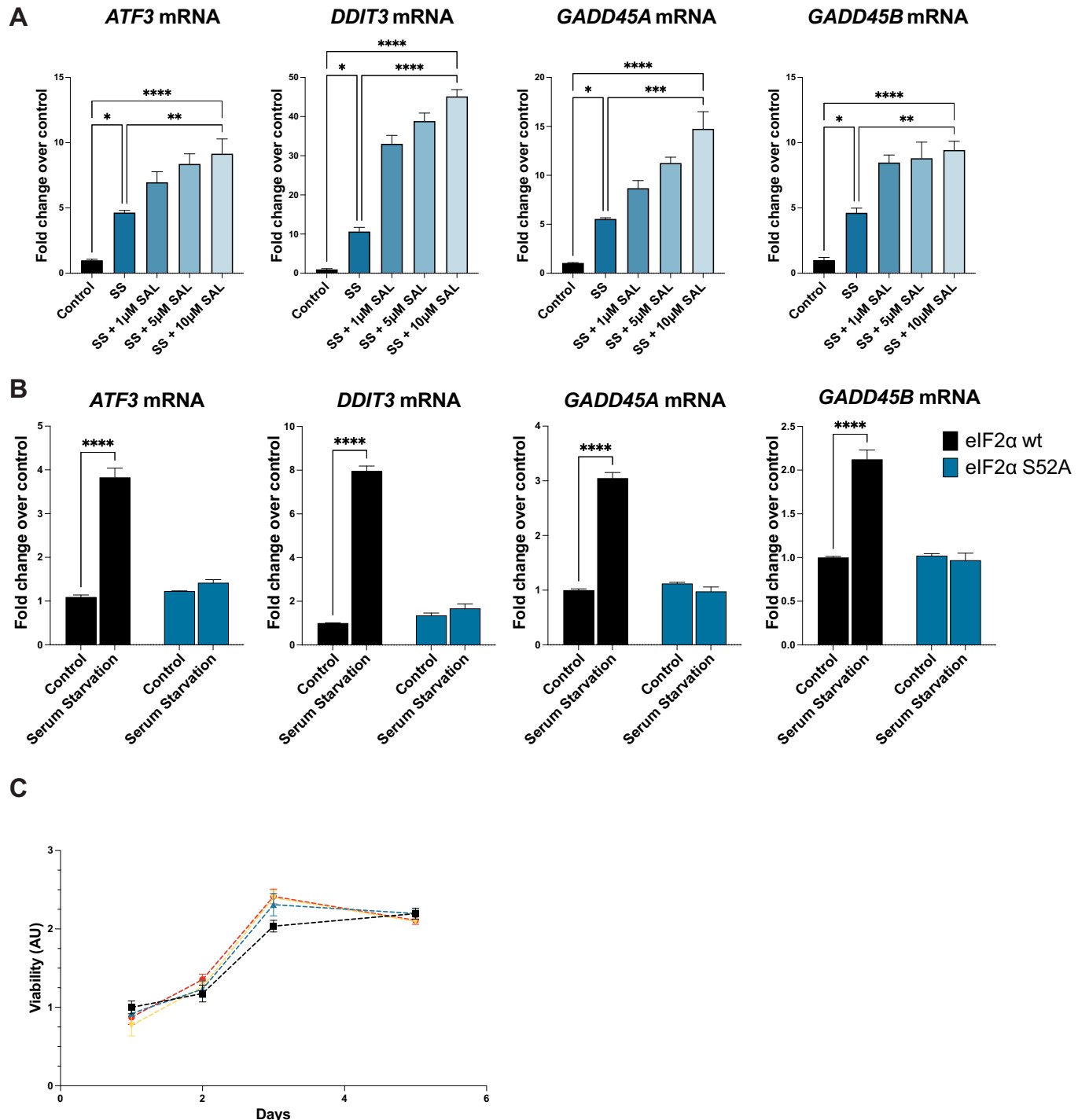

◄ **Figure EV4. MET expression and the ensuing biological activity are controlled by the *integrated stress response* regulator eIF2α.**

(A) RT-qPCR results showing the expression of ATF4 target genes (*ATF3, DDIT3, GADD45A,* and *GADD45B*) in EKVX cells treated with ± SS ± increasing amount of Salubrinal for 24 h. Data are represented as mean ± SEM with $n = 3$ biological triplicates and were analysed using one-way ANOVA with Tukey's post hoc test. (B) Bar graphs presenting the relative mRNA levels of *ATF3, DDIT3, GADD45A,* and *GADD45B* in cells expressing either MET wt or MET 5'UTR with eIF2α S52A mutation. The cells were subjected to ± serum starvation. Data are normalised to *TBP* expression levels and are represented as mean ± SEM ($n = 3$ biological replicates), and data are analysed by one-way ANOVA with post hoc tests comparing the effects of serum starvation on gene expression. Exact p values: ATF3 (Control vs. Serum Starvation, eIF2α wt: $P = 5.01 \times 10^{-5}$); DDIT3 (Control vs. Serum Starvation, eIF2α wt: $P = 3.02 \times 10^{-5}$); GADD45A (Control vs. Serum Starvation, eIF2α wt: $P = 2.16 \times 10^{-5}$); GADD45B (Control vs. Serum Starvation, eIF2α wt: $P = 9.31 \times 10^{-5}$). (C) Viability assay showing the growth curves of cells over six days under control conditions ±HGF, comparing wild-type with eIF2α S52A mutation. Data are represented as mean ± SEM with $n > 3$ biological replicates. Statistical test was performed using two-way ANOVA with repeated measures (non-significant). Significance marker: ****$P < 0.0001$, ***$P < 0.001$, **$P < 0.01$, and *$P < 0.05$.

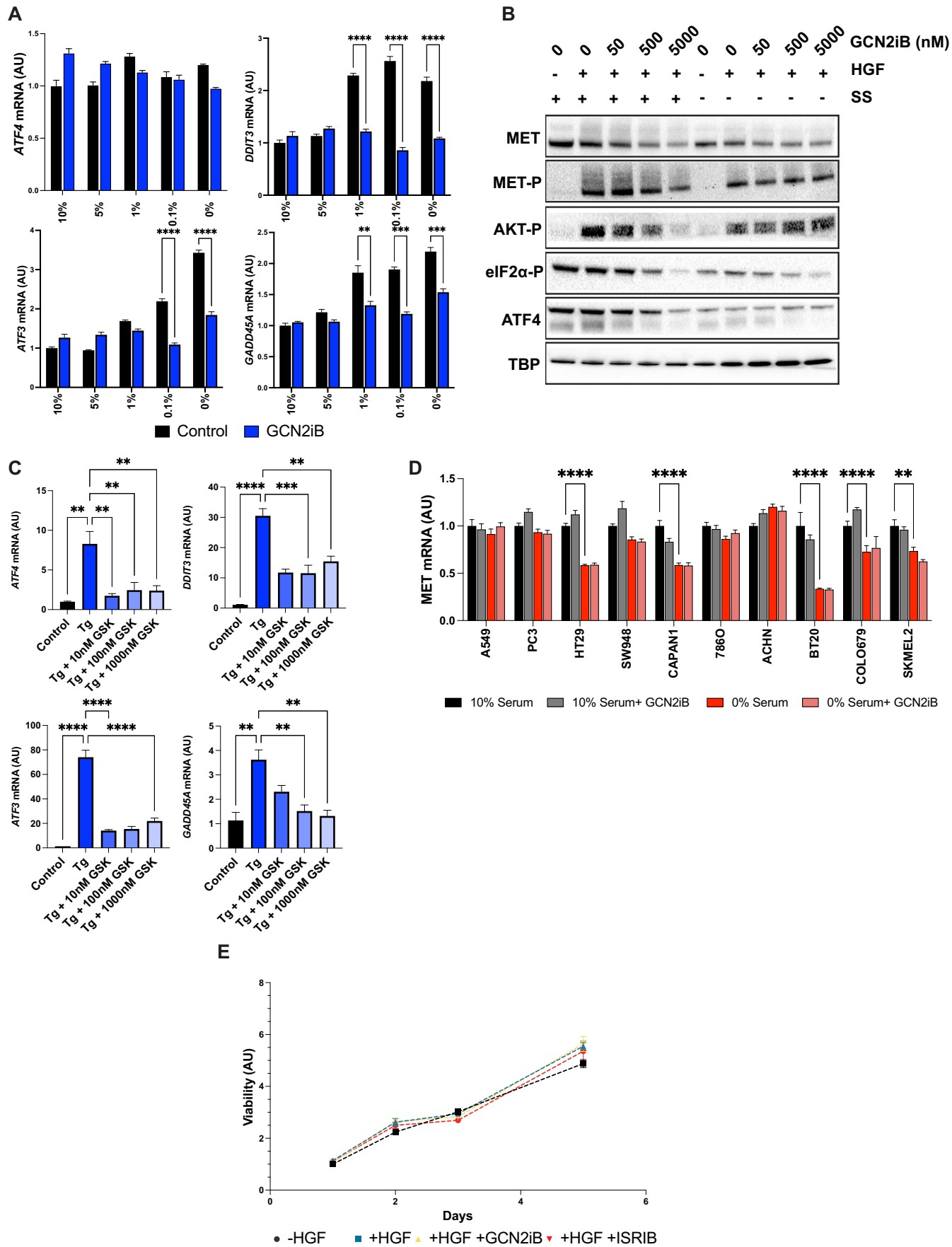

◀  **Figure EV5.  Stress-induced MET upregulation is impinged by ISR inhibition.**

(A) RT-qPCR analysis of *ATF4*, *DDIT3*, *ATF3*, and *GADD45A* in EKVX cells under control condition (10% serum) or decreasing amount of serum (from 5% to 0%) ± 500 nM GCN2iB. The graphs depict fold changes over control with *TBP* as a reference gene. Error bars represent SEM with $n = 3$ biological replicates. Data were analysed using two-way ANOVA. (B) Cells were cultured in 10% serum (control) or under serum starvation ± increasing concentrations of GCN2iB for 24 h and treated with ± 50 ng/mL of HGF for 15 min. Western blot panels showing MET, MET-P, AKT-P, eIF2α-P, and ATF4 levels, with TBP as the loading control. (C) RT-qPCR analysis of *ATF4*, *DDIT3*, *ATF3*, and *GADD45A* mRNA levels in EKVX cells treated with 500 nM Tg for 24 h in the presence or absence of the PERK inhibitor GSK2656157 (GSK, 10 nM, 100 nM, or 1000 nM). mRNA levels were normalised to *TBP*, and results are presented as fold changes over control conditions. Error bars represent SEM with $n = 3$ biological replicates. Data were analysed by one-way ANOVA with Tukey's post hoc test. (D) RT-qPCR experiments showing *MET* mRNA expression across a panel of cell lines under control condition (10% serum) or serum starvation (0% serum) ± 500 nM GCN2iB for 24 h. Results are fold change over control condition normalised with *TBP*. Results are presented as mean ± SEM with $n = 3$ biological replicate and analysed using two-way ANOVA. (E) The viability of EKVX cells is quantified over six days under control condition (10% Serum) ± HGF (50 ng/mL) ± ISR inhibitors GCN2iB or ISRIB. Error bars represent SEM with $n > 3$ biological replicates. Two-way ANOVA with repeated measures was used to evaluate significance (non-significant). Significance is denoted as ****$P < 0.0001$, ***$P < 0.001$, and **$P < 0.01$.

