## [Peer Review File · The EMBO Journal]

The integrated stress response drives MET oncogene overexpression in cancers

Marina Cerqua, Marco Foiani, Carla Boccaccio, Paolo Comoglio, and Dogus Altintas

Corresponding author(s): Dogus Altintas (dogus.altintas@ifom.eu) , Paolo Comoglio (paolo.comoglio@ifom.eu)

Review Timeline:

Submission Date:	17th May 24
Editorial Decision:	10th Jul 24
Revision Received:	23rd Sep 24
Editorial Decision:	21st Oct 24
Revision Received:	24th Oct 24
Editorial Decision:	7th Nov 24
Revision Received:	9th Nov 24
Accepted:	29th Nov 24

Editor: William Teale

Transaction Report:

Dear Dogus,

Thank you again for the submission of your manuscript entitled "The integrated stress response drives MET oncogene overexpression in cancers" (EMBOJ-2024-117914) and for your patience during the review process. We have now received the reports from the referees, which I copy below.

As you can see from their comments, while both referees detail important controls and propose accessible extensions to your hypotheses that should be addressed before your manuscript can be published in The EMBO Journal, both express a degree of enthusiasm for your work.

Based on the overall interest expressed in the reports, I would like to invite you to address the comments of all referees in a revised version of the manuscript. I should add that it is The EMBO Journal policy to allow only a single major round of revision and that it is therefore important to resolve the main concerns at this stage. I believe the concerns of the referees are reasonable and addressable, but please contact me if you have any questions, need further input on the referee comments or if you anticipate any problems in addressing any of their points. I suggest we have a Zoom call once you have had a chance to digest the reports; please let me know when would be a convenient time. Please also follow the instructions below when preparing your manuscript for resubmission.

I would also like to point out that as a matter of policy, competing manuscripts published during this period will not be taken into consideration in our assessment of the novelty presented by your study ("scooping" protection). We have extended this 'scooping protection policy' beyond the usual 3 month revision timeline to cover the period required for a full revision to address the essential experimental issues. Please contact me if you see a paper with related content published elsewhere to discuss the appropriate course of action.

Again, please contact me at any time during revision if you need any help or have further questions.

Thank you very much again for the opportunity to consider your work for publication. I look forward to your revision.

Best regards,

William

William Teale, Ph.D.
Editor
The EMBO Journal

When submitting your revised manuscript, please carefully review the instructions below and include the following items:

- 1) a .docx formatted version of the manuscript text (including legends for main figures, EV figures and tables). Please make sure that the changes are highlighted to be clearly visible.
- 2) individual production quality figure files as .eps, .tif, .jpg (one file per figure).
- 3) a .docx formatted letter INCLUDING the reviewers' reports and your detailed point-by-point response to their comments. As part of the EMBO Press transparent editorial process, the point-by-point response is part of the Review Process File (RPF), which will be published alongside your paper.
- 4) a complete author checklist, which you can download from our author guidelines ([https://wol-prod-cdn.literatumonline.com/pb-assets/embo-site/Author Checklist%20-%20EMBO%20J-1561436015657.xlsx](https://wol-prod-cdn.literatumonline.com/pb-assets/embo-site/Author%20Checklist%20-%20EMBO%20J-1561436015657.xlsx)). Please insert information in the checklist that is also reflected in the manuscript. The completed author checklist will also be part of the RPF.
- 5) Please note that all corresponding authors are required to supply an ORCID ID for their name upon submission of a revised manuscript.
- 6) We require a 'Data Availability' section after the Materials and Methods. Before submitting your revision, primary datasets produced in this study need to be deposited in an appropriate public database, and the accession numbers and database listed

under 'Data Availability'. Please remember to provide a reviewer password if the datasets are not yet public (see <https://www.embopress.org/page/journal/14602075/authorguide#datadeposition>). If no data deposition in external databases is needed for this paper, please then state in this section: This study includes no data deposited in external repositories. Note that the Data Availability Section is restricted to new primary data that are part of this study.

Note - All links should resolve to a page where the data can be accessed.

8) For data quantification: please specify the name of the statistical test used to generate error bars and P values, the number (n) of independent experiments (specify technical or biological replicates) underlying each data point and the test used to calculate p-values in each figure legend. The figure legends should contain a basic description of n, P and the test applied. Graphs must include a description of the bars and the error bars (s.d., s.e.m.).

9) We would also encourage you to include the source data for figure panels that show essential data. Numerical data can be provided as individual .xls or .csv files (including a tab describing the data). For 'blots' or microscopy, uncropped images should be submitted (using a zip archive or a single pdf per main figure if multiple images need to be supplied for one panel). Additional information on source data and instruction on how to label the files are available at .

10) We replaced Supplementary Information with Expanded View (EV) Figures and Tables that are collapsible/expandable online (see examples in <https://www.embopress.org/doi/10.15252/embj.201695874>). A maximum of 5 EV Figures can be typeset. EV Figures should be cited as 'Figure EV1, Figure EV2" etc. in the text and their respective legends should be included in the main text after the legends of regular figures.

12) Our journal encourages inclusion of *data citations in the reference list* to directly cite datasets that were re-used and obtained from public databases. Data citations in the article text are distinct from normal bibliographical citations and should directly link to the database records from which the data can be accessed. In the main text, data citations are formatted as follows: "Data ref: Smith et al, 2001" or "Data ref: NCBI Sequence Read Archive PRJNA342805, 2017". In the Reference list, data citations must be labeled with "[DATASET]". A data reference must provide the database name, accession number/identifiers and a resolvable link to the landing page from which the data can be accessed at the end of the reference. Further instructions are available at .

We realize that it is difficult to revise to a specific deadline. In the interest of protecting the conceptual advance provided by the work, we recommend a revision within 3 months (8th Oct 2024). Please discuss the revision progress ahead of this time with the editor if you require more time to complete the revisions. Use the link below to submit your revision:

Referee #1:

Cerqua et al. characterized and demonstrated the pivotal role of upstream ORFs located in the 5'UTR of the transcript encoding MET oncogene. Overall, they illustrated nicely the presence and functionality of ORFs in MET and showed their crucial roles in MET translation upon integrated stress response (ISR). Furthermore, considering MET involvement in epithelial mesenchymal transition (EMT), the authors did assess the impact of ISR on EMT upon MET stimulation with HGF. While ISR is proved to be involved in MET translation, further work needs to be addressed to clarify the mechanism, such as investigating the presence of IRES structures, and identify the key kinase(s) involved, notably.

Here the points that need to be addressed:

1- It is not clear from the text whether the 2 upstream ORFs were already annotated in database or if they were predicted by the authors. MET uORFs have been described in this Wethmar, et al., *Oncogene*, 2016. The text should include this reference.

2- uORFs and IRES can be found within the same 5'UTR sequence, and IRES can participate to the response to ISR. Did the authors try to predict IRES? Such prediction could include GC% and energy coefficient calculated with RNAfold prediction tools. If any stable structure is observed, putative IRES can be cloned in dual luciferase system for validation and in a promoterless vector as control.

3- Core experiments have been performed on EKVX lung adenocarcinoma cells. On western blot, MET doesn't appear well expressed. Why choosing this cell line?

4- The term 'housekeeping' is used several times in the manuscript to describe the behavior of MET upon rescue (uORFs mutation, eIF2 α S52A...). However, MET still doesn't behave like TBP, for example, used here like a housekeeping control on western blot. Therefore, the term 'housekeeping' needs to be changed by 'non stress regulated' or similar terms.

5- Gemcitabine is not a conventional drug to induce ISR, the authors used serum starvation which had also a broad impact on stress. ISR relies on the activation of 4 kinases that are activated by different stimuli: AA deprivation (GCN2), ER stress (PERK), viral infection (PKR), heme deprivation (HRI). Therefore, further analysis requires specific amino acid starvation such as glutamine, and ER stress inducers such as tunicamycin or thapsigargin.

6- On the same idea, different kinases inhibitors need to be tested, especially PERK inhibitor.

7- A current model to explain how uORFs work, involves re-initiation of the translation by the pre-initiation complex that would be de-phosphorylated directly on the transcript. ISRIB is known to stabilize the non-phospho pool of eIF2 α but doesn't promote its dephosphorylation. Therefore the "re-initiation" model can be tested here by using salubrinal which is an inhibitor of GADD34.

8- In Fig5, the results show here that ISR is important for EMT, and the implication of MET through HGF treatment doesn't prove the direct effect of MET. Results should be reproduced with MET mutated in its uORFs sequences in order to highlight the importance of uORFs in physiology.

Minor comments:

- WB Fig1F needs to be annotated to explain the double band
- Fig 2D, 4E and 5F don't have error bars
- Fig5 A needs to be re-organized according to - HGF and + HGF conditions, to facilitate the interpretation

Referee #2:

In this manuscript, Cerqua and colleagues showed a novel translational regulation of MET oncogene through the two upstream ORFs (uORFS) located in its 5'UTR that inhibit translation of the main oncogene. Upon activation of the integrated stress response (ISR) by different conditions such as serum starvation, hypoxia, irradiation, or gemcitabine treatment, phosphorylation of eIF2a increases and consequently increases protein levels of MET mRNA without altering its mRNA levels. This novel regulation is similar to the ATF4 regulation. I think the manuscript presents solid and exciting results for this regulation and will attract the general audience of EMBO Journal if the authors can address all the concerns listed below:

Main Concerns

- Please provide more information about two uORFs present in the 5' UTR of MET mRNA. How long are they? Are they both not in frame with the main ORF? How long peptides do they both produce? How similar are they compared to the uORFs present in ATF4?
- It is unclear why the authors preferred using a bicistronic reporter to study the role of uORFs. Using those reporters for cap-independent translation is more common in the field. Therefore, the mechanism proposed in this study does not require using those reporters. More importantly, it is essential to transfect in vitro transcribed RNA of the bicistronic reporters to avoid any cryptic splicing variants of the reporters. Therefore, it would be critical to use either mono cistronic reporters (Fluc and RLuc reporters, separately) or transfecting RNA of bicistronic reporters used in this manuscript to show the role of uORFs upstream of the main CDS. For the reporter assay, please provide the expression level of each reporter (normalized to No-5'UTR) in the control condition. What is the effect of the MET 5'UTR and its mutants on RLuc translation under normal conditions? On the same line, the uORF1/2 mut-GFP reporter is significantly more expressed than WT-5'UTR (Fig EV1B).
- It is unclear why MET protein levels decrease significantly after serum starvation in MET uORF1/2 mutated cells. The authors mention turning MET into a housekeeping gene, but in this case, why SS reduces its expression needs to be explained or discussed in the manuscript.
- It would be critical to use MET uORF1/2 mutated cells for wound healing assay in +/- HGF conditions to show that the mutation would decrease the effect of SS on cell migration.

Minor Concerns

- For Fig 1F and all the following panels, please indicate how long cells were kept under serum starvation conditions. It isn't clear to the reader as the effect is not always the same.
- Please explain the role of HGF and what to expect before showing the results in the text. It needs to be clarified for a general audience.
- Please provide IC50 values for Figures 2D, 3E, and 5F with SE/SD instead of showing bar graphs.
- The WT cells show different viability profiles between Fig 2D and 3E. Please discuss this discrepancy.
- It is important to provide a detailed introduction to the MET protein. The authors should explain why we should care about this oncogene and discuss its function/downstream pathways in the introduction.
- The MET mRNA levels for control cells are not comparable between Fig2A and 4B. Are they kept under different conditions? Why MET mRNA levels show a significant decrease in Fig 4B while not altered in Fig 2A?
- Are there any known mutations in the uORFs presented in this study that may affect the expression of the main ORF in any cancer? It would be interesting to at least discuss it.
- Statistical analyses should be clarified for those panels: Figures 1C, 1D, 5C, 5D.

Point-by-point responses to referees

Referee #1:

Cerqua et al. characterized and demonstrated the pivotal role of upstream ORFs located in the 5'UTR of the transcript encoding MET oncogene. Overall, they illustrated nicely the presence and functionality of ORFs in MET and showed their crucial roles in MET translation upon integrated stress response (ISR). Furthermore, considering MET involvement in epithelial mesenchymal transition (EMT), the authors did assess the impact of ISR on EMT upon MET stimulation with HGF. While ISR is proved to be involved in MET translation, further work needs to be addressed to clarify the mechanism, such as investigating the presence of IRES structures, and identify the key kinase(s) involved, notably. Here the points that need to be addressed:

1- It is not clear from the text whether the 2 upstream ORFs were already annotated in database or if they were predicted by the authors. MET uORFs have been described in this Wethmar, et al., *Oncogene*, 2016. The text should include this reference.

In the revised manuscript, we explicitly state that these uORFs were identified through analysis of existing databases. To clarify this, we have added two key references: Wethmar, et al., *Oncogene*, 2016, and Yang, et al., *Database*, 2021 (The Human IRES Atlas). These references have been incorporated into the Results section, where we discuss the similarities between the *MET* and *ATF4* 5'UTRs.

2- uORFs and IRES can be found within the same 5'UTR sequence, and IRES can participate to the response to ISR. Did the authors try to predict IRES? Such prediction could include GC% and energy coefficient calculated with RNAfold prediction tools. If any stable structure is observed, putative IRES can be cloned in dual luciferase system for validation and in a promoterless vector as control.

We appreciate the reviewer's insightful suggestion to explore the potential presence of an internal ribosome entry site (IRES) within the *MET* 5'UTR. To address this, we conducted computational predictions using the RNAfold Web Server and cross-referenced our findings with the Human IRES Atlas. The prediction suggested stable secondary structures between bases 302 and 396. We have included this structural prediction as part of Fig. EV2, providing a visual representation of the RNA folding and the predicted IRES.

Furthermore, following the reviewer's recommendation, we experimentally validated this putative IRES by cloning it into a dual-luciferase reporter system, placing the predicted IRES between Firefly and Renilla luciferase genes, with a promoterless vector as a control. These constructs were tested under various ISR-inducing conditions, including serum starvation, chemotherapy (Gemcitabine), irradiation, Thapsigargin-induced ER stress, Leucine deprivation, and hypoxia.

When transfected cells were challenged with these ISR stimuli, we observed no significant Firefly luciferase activity compared to the positive control (a known functional IRES), indicating that the predicted structure does not function as an IRES. Therefore, we conclude that *MET* upregulation under stress conditions is driven by the uORFs without involvement of IRES elements. These findings are presented in Fig. EV2D-E.

Notably, the absence of stress-induced IRES activity within the *MET* 5'UTR is consistent with results in Fig. 1C-D and Fig. 2: mutations in uORF1 and uORF2 either in a reporter system or by prime editing are sufficient to abolish the stress-responsiveness of *MET* translation.

3- Core experiments have been performed on EKVX lung adenocarcinoma cells. On western blot, MET doesn't appear well expressed. Why choosing this cell line?

We understand the reviewer's concern regarding the choice of the EKVX lung adenocarcinoma cell line for our core experiments. EKVX was selected for the following reasons:

- 1) **Cell Line Characteristics:** EKVX expresses wild-type, non-amplified *MET*, which allows us to study stress-induced *MET* expression in a model where *MET* is not overexpressed through genetic alterations, such as gene amplification. This makes EKVX ideal for focusing on the inducible nature of *MET* expression under stress while ensuring the ISR pathway remains intact for exploring its regulatory role.
- 2) **Established Model in Our Research:** EKVX is a well-characterized model in our research group, where it has been used extensively to study *MET*-induced invasive growth upon HGF treatment. Our familiarity with this cell line enables us to effectively track *MET* signalling and functional outcomes.
- 3) **Validation in Additional Cell Lines:** To ensure our findings are not specific to EKVX, we have validated ISR-dependent *MET* expression in ten additional cell lines, all expressing *MET* at levels comparable to EKVX. These results, presented in Fig. 4G-I and Fig. EV5D, confirm the generality of our findings across multiple models.

We believe that the use of the EKVX cell line, combined with validation in additional models, provides a robust and reliable basis for the conclusions drawn in our study.

4- The term 'housekeeping' is used several times in the manuscript to describe the behavior of *MET* upon rescue (*uORFs* mutation, *eIF2 α* S52A...). However, *MET* still doesn't behave like TBP, for example, used here like a housekeeping control on western blot. Therefore, the term 'housekeeping' needs to be changed by 'non stress regulated' or similar terms.

We agree that the term 'housekeeping' may not be fully appropriate in this context. We have revised the manuscript to replace the term 'housekeeping' with 'non-stress regulated' as suggested. This change has been made throughout the manuscript to more accurately describe the behaviour of the genes in question.

5- Gemcitabine is not a conventional drug to induce ISR, the authors used serum starvation which had also a broad impact on stress. ISR relies on the activation of 4 kinases that are activated by different stimuli: AA deprivation (*GCN2*), ER stress (*PERK*), viral infection (*PKR*), heme deprivation (*HRI*). Therefore, further analysis requires specific amino acid starvation such as glutamine, and ER stress inducers such as thapsigargin or thapsigargin.

We appreciate the reviewer's insightful suggestions. While Gemcitabine (GEM) has been shown to induce ISR, we agree that using more conventional ISR-inducing agents would provide clearer insights. Following the reviewer's recommendations, we have now incorporated Thapsigargin treatments and Leucine deprivation into our experiments. These treatments were applied in both reporter assays and Western blot analyses (Fig. 1 C-E). Our results show that the removal of a single amino acid (L-Leucine) or ER stress induced by Thapsigargin effectively triggered ISR and increased *MET* protein levels. These additional findings are presented in the revised manuscript.

6- On the same idea, different kinases inhibitors need to be tested, especially PERK inhibitor.

In response to the reviewer's suggestion, we tested the effect of a PERK inhibitor to further validate our findings. These experiments have been incorporated into Fig. 4C-D and Fig. EV5C. The results demonstrate that inhibition of PERK effectively blocked ISR-induced MET upregulation, further confirming the role of PERK in the regulation of MET expression under stress. We believe that these additional data strengthen our study and align well with the reviewer's recommendations.

7- A current model to explain how uORFs work, involves re-initiation of the translation by the pre-initiation complex that would be de-phosphorylated directly on the transcript. ISRIB is known to stabilize the non-phospho pool of eIF2 α but doesn't promote its dephosphorylation. Therefore the "re-initiation" model can be tested here by using salubrinal which is an inhibitor of GADD34.

Thank you for this insightful suggestion. To investigate the role of the re-initiation mechanism in uORF-mediated regulation of *MET* translation, we conducted additional experiments using Salubrinal, an inhibitor of GADD34, which prevents eIF2 α dephosphorylation. These experiments aimed to clarify how the phosphorylation state of eIF2 α influences *MET* expression.

In these new experiments, EKVX lung adenocarcinoma cells were serum-starved to induce the Integrated Stress Response (ISR). Salubrinal was applied at varying concentrations (1 μ M, 5 μ M, and 10 μ M) to inhibit eIF2 α dephosphorylation. Salubrinal treatment resulted in sustained eIF2 α phosphorylation, accompanied by a modest but consistent increase in ATF4 and *MET* protein levels (Fig. 3C), as well as ATF4 target gene expression (Fig. EV4A), which aligns with the re-initiation model.

8- In Fig5, the results show here that ISR is important for EMT, and the implication of *MET* through HGF treatment doesn't prove the direct effect of *MET*. Results should be reproduced with *MET* mutated in its uORFs sequences in order to highlight the importance of uORFs in physiology.

Thank you for the valuable suggestion. To directly assess the impact of uORF mutations on *MET* expression and EMT, we conducted additional experiments. As shown in Figure 2A, there is a clear differential regulation of *MET* expression between wild-type and uORF1/2-mutated *MET* under both steady-state and serum starvation (SS) conditions.

Steady State: In the absence of stress, *MET* expression is elevated in cells with uORF1/2 mutations compared to WT cells. This increased *MET* expression is associated with enhanced activation of downstream signalling pathways, including AKT phosphorylation (AKT-P). Consequently, higher levels of EMT gene expression are observed in these mutated cells (Fig. 2C).

Serum Starvation (SS): Under SS conditions, in WT cells, *MET* protein levels are high, with strong HGF-driven signalling (indicated by *MET*-P and AKT-P), driving EMT processes. In contrast, uORF1/2-mutant cells display reduced *MET* expression and signalling, leading to decreased EMT gene expression (Fig. 2C).

These results highlight the critical role of uORFs in modulating *MET* expression and its downstream effects on EMT.

Minor comments:

- WB Fig1F needs to be annotated to explain the double band

The figure has been annotated to highlight the precursor (P) and the mature (M) form of MET.

- Fig 2D, 4E and 5F don't have error bars

We have added an inset presenting the IC50 values along with error bars for the data shown in the new Fig. 2E, 3G, and 5F.

Regarding Fig. 4E (current version, Fig. 4H), which includes data from the Cancer Cell Line Encyclopedia (CCLE), we would like to clarify the absence of error bars. The CCLE provides normalized gene expression data for various cell lines, typically presented as single values per cell line without associated error bars. This is because the CCLE data represent aggregated expression levels from large-scale sequencing or microarray experiments across many cell lines. Therefore, this figure serves as a comparative illustration rather than a presentation of experimental variability, making error bars not applicable in this context.

- Fig5 A needs to be re-organized according to - HGF and + HGF conditions, to facilitate the interpretation

We appreciate the reviewer's suggestion to reorganize Fig. 5A according to the presence or absence of HGF to facilitate interpretation. We have made the requested changes, and the figure has been reorganized to differentiate between the +HGF and -HGF conditions.

Referee #2:

In this manuscript, Cerqua and colleagues showed a novel translational regulation of MET oncogene through the two upstream ORFs (uORFs) located in its 5'UTR that inhibit translation of the main oncogene. Upon activation of the integrated stress response (ISR) by different conditions such as serum starvation, hypoxia, irradiation, or gemcitabine treatment, phosphorylation of eIF2a increases and consequently increases protein levels of MET mRNA without altering its mRNA levels. This novel regulation is similar to the ATF4 regulation. I think the manuscript presents solid and exciting results for this regulation and will attract the general audience of EMBO Journal if the authors can address all the concerns listed below:

Main Concerns

- Please provide more information about two uORFs present in the 5' UTR of MET mRNA. How long are they? Are they both not in frame with the main ORF? How long peptides do they both produce? How similar are they compared to the uORFs present in ATF4?

To address this feedback, we included the requested information in the Results section of the revised manuscript. Specifically, we now provide details on the two uORFs present in the *MET* 5' UTR, including their lengths, whether they are in-frame or out-of-frame with the main ORF, and the lengths of the peptides they might produce. Additionally, we have compared these uORFs to those in *ATF4*, highlighting their similarities and differences to further clarify their roles in translation regulation.

- It is unclear why the authors preferred using a bicistronic reporter to study the role of uORFs. Using those reporters for cap-independent translation is more common in the field. Therefore, the mechanism proposed in this study does not require using those reporters. More importantly, it is essential to transfect in vitro transcribed RNA of the bicistronic reporters to avoid any cryptic splicing variants of the reporters. Therefore, it would be critical to use either mono cistronic reporters (Fluc and RLuc reporters, separately) or transfecting RNA of bicistronic reporters used in this manuscript to show the role of uORFs upstream of the main CDS. For the reporter assay, please provide the expression level of each reporter (normalized to No-5'UTR) in the control condition. What is the effect of the MET 5'UTR and its mutants on Rluc translation under normal conditions? On the same line, the uORF1/2 mut-GFP reporter is significantly more expressed than WT-5'UTR (Fig EV1B).

In response to the reviewer's concerns, we have transitioned to monocistronic vectors to avoid potential confounding effects and to directly assess the impact of uORFs on reporter gene expression. This modification provides a clearer interpretation of the specific role of uORFs. Importantly, the results obtained with monocistronic vectors are consistent with our previous findings using bicistronic vectors, and these findings are further supported by an additional reporter gene, GFP, as presented in Fig. EV1. In the revised manuscript, we have now included the expression levels of each reporter, normalized to the No-5'UTR control.

Moreover, we have addressed the reviewer's concern regarding the GFP reporter system (Fig. EV1B-C), where the uORF1/2 mutant GFP reporter is significantly more expressed than the wt-5'UTR. These findings were validated using monocistronic assays, confirming the impact of uORF mutations on translation efficiency. Mutating both uORFs of the *MET* 5'UTR enhances reporter gene expression under steady-state conditions, aligning with the known inhibitory role of uORFs. These observations are

consistent with the higher endogenous MET levels observed in uORF1/2 mutants generated via prime editing under steady-state conditions (Fig. 2A).

- It is unclear why MET protein levels decrease significantly after serum starvation in MET uORF1/2 mutated cells. The authors mention turning MET into a housekeeping gene, but in this case, why SS reduces its expression needs to be explained or discussed in the manuscript.

We have clarified this point in the revised manuscript. During ISR activation, global protein synthesis is generally reduced, while transcripts containing functional uORFs are preferentially translated. When the uORFs within the MET 5'UTR are mutated, this regulatory mechanism is disrupted. This explains the significant decrease in MET protein levels in uORF1/2 mutated cells. We have included this explanation in the Results section of the revised manuscript.

- It would be critical to use MET uORF1/2 mutated cells for wound healing assay in +/- HGF conditions to show that the mutation would decrease the effect of SS on cell migration.

Thank you for this insightful suggestion. We recognize the importance of assessing cell migration in MET uORF1/2-mutated cells under +/- HGF conditions. However, the wound healing assay, which requires serum removal just before HGF treatment, yielded counterintuitive results. uORF1/2-mutated cells migrated significantly faster than WT cells, likely due to the higher MET expression at a steady state (Fig. 2A and wound healing data, attached below). Longer pre-incubation in a serum-free medium completely inhibited migration, rendering this assay unsuitable.

To better illustrate the importance of uORFs in MET physiology, we focused on epithelial-mesenchymal transition (EMT), another key feature of MET function. In the revised manuscript, we have included a new figure demonstrating the impact of uORF1/2 mutations on MET-driven EMT under \pm serum and \pm HGF conditions (Fig. 2C).

Minor Concerns

- For Fig 1F and all the following panels, please indicate how long cells were kept under serum starvation conditions. It isn't clear to the reader as the effect is not always the same.

We have added the duration of serum starvation for the cells in all relevant panels in the revised manuscript.

- Please explain the role of HGF and what to expect before showing the results in the text. It needs to be clarified for a general audience.

In the revised manuscript, we have expanded the Introduction to recall the signalling pathways triggered by HGF upon binding to the MET receptor.

- Please provide IC50 values for Figures 2D, 3E, and 5F with SE/SD instead of showing bar graphs.

Thank you for the suggestion. We have provided the IC50 values for Figures 2D (now Fig. 2E), 3E (now Fig. 3G), and 5F, along with their respective standard error (SE), as recommended. This additional information has been included in the revised manuscript.

- The WT cells show different viability profiles between Fig 2D and 3E. Please discuss this discrepancy.

Thank you for highlighting this point. We carefully re-analyzed the IC50 values for WT cells presented in Fig. 2D (now Fig. 2E, Exp 1) and Fig. 3E (now Fig. 3G, Exp 2). As shown in the accompanying bar plot, the IC50 values for WT cells are comparable between the two experiments, with no significant differences, as indicated by the "ns" (non-significant) markers. This consistency confirms that there is no actual discrepancy in WT cell viability between Fig. 2D and Fig. 3E, as the observed IC50 values fall within a similar range across independent experiments.

- It is important to provide a detailed introduction to the MET protein. The authors should explain why we should care about this oncogene and discuss its function/downstream pathways in the introduction.

In the revised manuscript, we have expanded the Introduction to provide a concise overview of the *MET* oncogene with references to key reviews. We also emphasize its pervasive overexpression in cancer, occurring in a wide spectrum of cases, crowning *MET* as one of the top five oncogenes to prioritise for targeted therapies (Behan, et al., Nature, 2019). This underscores its relevance in cancer research and highlights the critical importance of understanding its regulation.

- The MET mRNA levels for control cells are not comparable between Fig2A and 4B. Are they kept under different conditions? Why MET mRNA levels show a significant decrease in Fig 4B while not altered in Fig 2A?

The differences in *MET* mRNA levels between Fig 2B and Fig 4B are likely due to variations in the passage number of the cells, which can occur during the CRISPR/Cas9 protocol. Passage number is known to influence gene expression (Yang, Y.-H.K., et al., Stem Cell Research & Therapy, 2018; Esquenet,

M., et al., J Steroid Biochem Mol Biol, 1997). While this aspect of transcriptional variability is interesting, it is beyond the focus of our current study.

- Are there any known mutations in the uORFs presented in this study that may affect the expression of the main ORF in any cancer? It would be interesting to at least discuss it.

We acknowledge the reviewer for raising this important point. The potential impact of mutations in uORFs on the expression of the main ORF in various cancers is indeed a fascinating area of research. The investigation of such mutations and their effects on cancer are beyond the scope of the current study. We recognize the significance of this question and are actively exploring it in ongoing research.

- Statistical analyses should be clarified for those panels: Figures 1C, 1D, 5C, 5D.

Thank you for your comment. We have clarified the statistical analyses in the figure legends for Figures 1C and 1D. Specifically, p-values were calculated using two-way ANOVA with Tukey's post-hoc test, representing the row factor (different vectors) and column factor (different treatments).

For Figures 5C and 5D, we used a mixed-effects model to analyze the data, accounting for both fixed effects (time and treatment) and random effects (subject variability). P-values were calculated for time, treatment conditions (column factor), and their interaction, all of which were statistically significant.

These statistical approaches have been clarified in the figure legends and the Materials and Methods section.

Dear Dogus,

Thank you for submitting your manuscript for consideration by the EMBO Journal. It has now been seen by three referees whose comments are shown below.

Whilst Referee #2 is satisfied with the revisions you have made, Referee #1 requests (in addition to a more detailed description of the statistical tests you applied) a final complementation experiment. I agree that this appears an easily-achievable and yet potentially very valuable final test of your hypothesis. I would like to hear your thoughts on the feasibility of this work, either by email, or Zoom call if you prefer. Please let me know how you would like to proceed.

Bets wishes,

William

William Teale, PhD
Editor
The EMBO Journal
w.teale@embojournal.org

We realize that it is difficult to revise to a specific deadline. In the interest of protecting the conceptual advance provided by the work, we recommend a revision within 3 months (19th Jan 2025). Please discuss the revision progress ahead of this time with the editor if you require more time to complete the revisions. Use the link below to submit your revision:

Referee #1:

The authors have addressed successfully most of the comments.

A remaining concern is that many experiments lack statistical analyses, despite errors bars are displayed. Statistical analyses must therefore be shown and when necessary the number of independent experimental repeats indicated.

Another concern is raised in Fig5, the results show the importance of ISR and HGF in EMT, however the role of uORF in the phenotype is not clearly elucidated. This could be tested by a rescue experiment with mutated uORF MET transcripts. If the authors are unable to achieve such an experiment, conclusions (discussion, lines 340-343 in particular) should be toned down accordingly.

Minor Comments :

Line 50 - cite the two other ISR kinases : HRI and PKR

Line 78 -80 (HIF and NF-kB only explain partially the overexpression of MET) : references are missing

Line 116 - Fig 1C and 1D. Calculate Ratio Firefly/Renilla and display statistical analysis on the figure.

Line 125 - Replace "To its functionality" with "To test its functionality"

Line 127 - Which IRES is used as a control. Is it a viral/mammalian IRES ?

Line 129 - Fig EV2D and E. Calculate Ratio Firefly/Renilla and display statistical analysis on the figure.

Line 170-176 - References or data are missing to support the text

Line 214 - MET phosphorylation reduction accompanies total MET downregulation. Authors cannot conclude directly to a lower MET phosphorylation.

Line 334 - "passive ISR" is not a known concept. This should be explained or modified.

Referee #2:

The authors have thoroughly addressed all the concerns and comments raised in the initial review. The revisions have strengthened the paper significantly, and I believe it now meets the standards for publication in the EMBO Journal. I recommend that the manuscript be accepted.

Referee #1:

The authors have addressed successfully most of the comments.

Thank you very much for your constructive feedback and the time you have taken to review our work.

A remaining concern is that many experiments lack statistical analyses, despite errors bars are displayed. Statistical analyses must therefore be shown and when necessary the number of independent experimental repeats indicated.

We have now ensured that statistical analyses and the number of independent experiments are included in all relevant figure panels.

Another concern is raised in Fig5, the results show the importance of ISR and HGF in EMT, however the role of uORF in the phenotype is not clearly elucidated. This could be tested by a rescue experiment with mutated uORF MET transcripts. If the authors are unable to achieve such an experiment, conclusions (discussion, lines 340-343 in particular) should be toned down accordingly.

Thank you for highlighting this important point. The experiment testing the role of uORF in the EMT phenotype was indeed performed, and we fully agree that it is a key experiment. Therefore, we placed this data in Figure 2C rather than Figure 5, as we felt it was important to show the biological relevance of the uORFs earlier in the manuscript to clarify their role in HGF-triggered EMT.

We apologize if this was not clear in the original manuscript. To ensure the data is more accessible and highlights the role of uORFs, we have rewritten the relevant section of the Results to improve clarity and emphasize these observations (Lines 170-181). Additionally, following your suggestion, we have moderated our conclusions in Lines 360-364 to avoid any potential overstatement.

Minor Comments :

Line 50 - cite the two other ISR kinases : HRI and PKR

We have cited the two additional ISR kinases, HRI and PKR, as requested (Line 50).

Line 78 -80 (HIF and NF- κ B only explain partially the overexpression of MET) : references are missing

Missing references for the role of HIF and NF- κ B in partially explaining MET overexpression have now been added (Lines 84-85).

Line 116 - Fig 1C and 1D. Calculate Ratio Firefly/Renilla and display statistical analysis on the figure.

"For the reporter assay, please provide the expression level of each reporter (normalized to No-5'UTR) in the control condition." Reviewer 2 question in revision round 1. So we prefer keeping in separate format. However, statistical significances are clearly shown in figure 1C.

We appreciate your suggestion to calculate and display the Firefly/Renilla ratios. In the previous round of revisions, Referee #2 requested that we provide the expression level of each reporter separately (normalised to the No-5'UTR condition). To address both comments, we opted to present the data separately to clearly display the individual expression levels, which we believe adds clarity to the interpretation of the data.

We have ensured that the statistical significance for all conditions is clearly displayed in Figure 1C-D, as requested so that the reader can readily assess the differences between groups.

We hope this approach effectively balances both reviewers' suggestions and preserves the clarity of the data presentation.

Line 125 - Replace "To its functionality" with "To test its functionality"

We have corrected the phrase "To its functionality" to "To test its functionality" (Line 128)

Line 127 - Which IRES is used as a control. Is it a viral/mammalian IRES ?

We have specified the type of IRES used as a control in the bicistronic assay as a viral IRES (Poliovirus IRES, Line 130).

Line 129 - Fig EV2D and E. Calculate Ratio Firefly/Renilla and display statistical analysis on the figure.

To maintain consistency with the format used in Figures 1C and 1D, we have opted to display the Renilla and Firefly luciferase data separately in Figures EV2D and EV2E. We believe that presenting the data this way allows for a clearer interpretation of the individual reporter activities.

That said, we have ensured that statistical tests are now clearly shown in Figures EV2D and EV2E, as per your suggestion.

We hope this format addresses your concerns while keeping the data presentation consistent across the manuscript.

Line 170-176 - References or data are missing to support the text

Thank you for pointing this out. We have now ensured that the relevant data is clearly explained and directly referenced within the figures. The text has been revised to better align with the corresponding figures that substantiate the statements made, with appropriate figure citations now included to support the data (Lines 175-181).

Line 214 - MET phosphorylation reduction accompanies total MET downregulation. Authors cannot conclude directly to a lower MET phosphorylation.

In response to the comment regarding MET phosphorylation, we have reworded the text. The new sentence reads: "Consistent with the lower MET protein levels in S52A cells, reduced HGF-induced MET phosphorylation and downstream AKT activation were observed compared to wild-type cells (Fig. 3D)" (Lines 227-229).

Line 334 - "passive ISR" is not a known concept. This should be explained or modified.

The phrase "passive ISR" has been reformulated for clarity, as it is not a recognized concept. The corrected version reads: "Far from being a passive marker of ISR activation, our findings underscore MET's active participation in the cellular response to environmental challenges." (Lines 356-357).

Referee #2:

The authors have thoroughly addressed all the concerns and comments raised in the initial review. The revisions have strengthened the paper significantly, and I believe it now meets the standards for publication in the EMBO Journal. I recommend that the manuscript be accepted.

Thank you very much for your positive feedback and for acknowledging the revisions made to our manuscript. We greatly appreciate your thorough review and are pleased that the changes have strengthened the paper to meet the standards of the EMBO Journal.

Dear Dogus,

Thank you for submitting the revised version of your manuscript, which addresses the concerns of the referees. This revised version has now been re-reviewed; I attach the second referee reports to the bottom of this mail. As you will see, you have addressed the referees' concerns to their satisfaction. Reviewer 1 makes some final constructive suggestions which I would like you to consider carefully. Before I can finally accept the manuscript, there are some remaining editorial points which need to be addressed. In this regard, would you please:

- acknowledge grant number 4691 (for Fondazione Umberto Veronesi) in the manuscript file,
- include up to five keywords,
- remove the author credit section from the manuscript file,
- remove callout in the manuscript text for figure 2,
- remove the reagents and tools section from the manuscript and upload as an individual file (For more information, please check <https://www.embopress.org/page/journal/14602075/authorguide#structuredmethods> and download the template for Reagent Table) ,
- save source data files in a scheme of one figure per folder and then upload as .zip files. (for example all the source data files for figure 1 need to be saved in a single folder and this needs to be zipped and then uploaded as "SD figure 1.zip" file),
- define the annotated p values ****/**/*/* as well as provide the exact p-values for the same in the legends of figures 1c; 5b-d; EV 1c; EV 3a; EV 4b,
- indicate the statistical test used for data analysis in the legends of figures 2e; 3g; 5f; EV 4a,
- define n in the legends of figures 2b; 3e-g; 4b; 5b, d, f; EV 1c; EV 2d-e; EV 3a-b; EV 4a-c; EV 5a, c-e,
- describe the nature of entity for 'n' in the legend of figure 5c,
- define error bars in the legends of figures 3f-g; 4b; EV 2d-e; EV 3b; EV 4a, c; EV 5e,
- define the measure of centre for the error bars in the legends of figures 1c; 2d-e,
- provide a numbered scale bar for heatmap in figure 1e,
- provide a scale bar for figure EV 1b, and
- correct the section order as follows: title page with complete author information, abstract, keywords, introduction, results, discussion, methods, data availability section, acknowledgements, disclosure and competing interests statement, references, main figure legends, tables, expanded figure legends.

Press is an editorially independent publishing platform for the development of EMBO scientific publications.

Best wishes,

William

William Teale, PhD
Editor
The EMBO Journal
w.teale@embojournal.org

We realize that it is difficult to revise to a specific deadline. In the interest of protecting the conceptual advance provided by the work, we recommend a revision within 3 months (5th Feb 2025). Please discuss the revision progress ahead of this time with the editor if you require more time to complete the revisions. Use the link below to submit your revision:

All editorial and formatting issues were resolved by the authors.

Dear Dogus,

I am pleased to inform you that your manuscript has been accepted for publication in the EMBO Journal.

Congratulations!

Best wishes,

William

William Teale, PhD
Editor
The EMBO Journal
w.teale@embojournal.org
